# Acetylation reprograms MITF target selectivity and residence time

Pakavarin Louphrasitthiphol [1,2], Alessia Loffreda [3], Vivian Pogenberg [4,12], Sarah Picaud[5], Alexander Schepsky[1,6], Hans Friedrichsen[1], Zhiqiang Zeng[7], Anahita Lashgari[8], Benjamin Thomas[9], E. Elizabeth Patton [7], Matthias Wilmanns[4,10], Panagis Filippakopoulos[5], Jean-Philippe Lambert [8], Eiríkur Steingrímsson[6], Davide Mazza[3,11] & Colin R. Goding [1] ✉

The ability of transcription factors to discriminate between different classes of binding sites associated with specific biological functions underpins effective gene regulation in development and homeostasis. How this is achieved is poorly understood. The microphthalmia-associated transcription factor MITF is a lineage-survival oncogene that plays a crucial role in melanocyte development and melanoma. MITF suppresses invasion, reprograms metabolism and promotes both proliferation and differentiation. How MITF distinguishes between differentiation and proliferation-associated targets is unknown. Here we show that compared to many transcription factors MITF exhibits a very long residence time which is reduced by p300/CBP-mediated MITF acetylation at K206. While K206 acetylation also decreases genome-wide MITF DNA-binding affinity, it preferentially directs DNA binding away from differentiation-associated CATGTG motifs toward CACGTG elements. The results reveal an acetylation-mediated switch that suppresses differentiation and provides a mechanistic explanation of why a human K206Q MITF mutation is associated with Waardenburg syndrome.

The gene expression programs that underpin development and homeostasis are mediated by the activity of sequence specific transcription factors that up- or down-regulate their target genes in response to intra- and extracellular cues. De-regulation of signals that are interpreted by transcription factors can lead to altered gene expression programs associated with diseases such as cancer. The ability of any given transcription factor to find its target elements in the genomic context is affected by several factors including the affinity of the transcription factor for DNA, its abundance, and whether its cognate recognition element is accessible or occluded by nucleosomes[1-3]. Importantly, the same transcription factor may regulate different sets of target genes with fundamentally different biological functions, implying that their targeting may be regulated. How transcription factors distinguish between different repertoires

[1]Ludwig Institute for Cancer Research, Nuffield Department of Clinical Medicine, University of Oxford, Headington, Oxford, UK. [2]Department of Gastro-intestinal and Hepato-Biliary-Pancreatic Surgery, Faculty of Medicine, University of Tsukuba, Ibaraki, Japan. [3]Experimental Imaging Center, Ospedale San Raffaele, Milano, Italy. [4]European Molecular Biology Laboratory, Hamburg Unit, Hamburg, Germany. [5]Structural Genomics Consortium, Nuffield Department of Clinical Medicine, University of Oxford, Headington, Oxford, UK. [6]Department of Biochemistry and Molecular Biology, BioMedical Center, Faculty of Medicine, University of Iceland, Reykjavik, Iceland. [7]MRC Institute of Genetics and Molecular Medicine, MRC Human Genetics Unit & Edinburgh Cancer Research Centre, Edinburgh, UK. [8]Department of Molecular Medicine and Cancer Research Center, Université Laval, Quebec, Canada; Endocrinology – Nephrology Axis, CHU de Québec – Université Laval Research Center, Quebec City, QC, Canada. [9]Central Proteomics Facility, Sir William Dunn Pathology School, University of Oxford, Oxford, UK. [10]University Hamburg Medical Centre Hamburg-Eppendorf, Hamburg, Germany. [11]Università Vita-Salute San Raffaele, Milano, Italy. [12]Present address: Institute of Biochemistry and Signal Transduction, University Hamburg Medical Centre Hamburg-Eppendorf, Hamburg, Germany. ✉e-mail: colin.goding@ludwig.ox.ac.uk

of binding sites in response to cell extrinsic or intrinsic cues is poorly understood.

The microphthalmia-associated transcription factor MITF[4] plays a critical role in development of the melanocyte lineage where it controls survival of melanoblasts[5]. Consequently reduced expression or mutation can lead to Waardenburg syndrome, characterized by pigmentation defects[6]. MITF has a key coordinating role in regulating many cellular functions, and in melanoma, a highly aggressive skin cancer originating in melanocytes, MITF has been defined as a lineage survival oncogene[7]. In addition to up-regulating a repertoire of melanocyte differentiation-associated genes, MITF suppresses invasion and senescence, drives proliferation, lysosome biogenesis and autophagy, and reprograms metabolism[8–22]. To account for its apparently contradictory roles in promoting both differentiation and proliferation, a rheostat model was proposed for MITF function in which low levels of MITF activity are associated with invasion, intermediate activity with proliferation, and high activity drives differentiation[10]. Yet while this model is a useful approximation to explain MITF's role in cell biology[23], it does not explain how differentiation is prevented in proliferating cells. One clue may be provided by the target sequence specificity of MITF. As a basic-helix-loop-helix-leucine zipper (bHLH-LZ) transcription factor MITF recognizes 6 bp E-box motifs, with 5′ flanking T residues that facilitate recognition by MITF and prevent binding by MYC[24–28]. MITF recognizes both core CACGTG and CATGTG E-box elements, with the latter being found in differentiation-associated genes such as *TYR* and *ABCB5*[29–31]. Whether or how MITF might differentiate between these two classes of target site is unknown, but is important since preventing binding to CATGTG elements could in principle lead to de-differentiation. One plausible model is that stable recognition of differentiation genes that contain CATGTG M-box motifs, but not CACGTG E-boxes, is prevented by MITF post-translational modification of residues implicated in direct DNA binding, though to date no such MITF modification has been identified.

In melanoma, the major driver mutations in BRAF and NRAS lead to activation of the MAPK pathway, and ERK-mediated phosphorylation of MITF has been variously reported to control its nuclear export[32], stability[33] and recruitment of the acetyl transferases CBP/p300[34]. Moreover, increased pro-proliferative MAPK signaling increases acetylation by promoting p300/CBP activity[35]. Although it remains unclear if or how deregulated MAPK signaling might enable MITF to differentiate between its targets implicated in proliferation versus those driving differentiation, understanding how MITF activity is regulated is especially important in melanoma as varying MITF activity has been associated with drug and immunotherapy resistance[14–17,36–40].

Here we reveal that compared to many transcription factors MITF exhibits a very long residence time which is decreased by acetylation at MITF K206. Notably acetyl K206 preferentially decreases MITF DNA-binding affinity for differentiation-associated CATGTG elements and shifts the equilibrium of MITF binding towards CACGTG motifs. The results highlight a mechanism by which activation of MAPK signaling would prevent melanocyte differentiation, and may also explain why a human K206Q mutation leads to Waardenburg syndrome.

## Results

Like all bHLH-LZ factors, DNA-binding by MITF is mediated by a basic region that makes base-specific contacts that determine target sequence specificity, as well as non-specific contacts mediated by basic amino acid interaction with the phosphate backbone[41]. In considering the possible mechanisms by which bHLH-LZ factor target gene selectivity could be regulated, we hypothesized that a post-translational modification affecting amino acids directly contacting DNA could be important in either changing MITF target specificity directly, or by affecting MITF's DNA binding affinity and indirectly modulating its capacity to bind selectively different elements.

Consistent with previous observations that MITF can be acetylated by the p300 and CBP acetyl transferases[29], well-established MAPK-activated cofactors for MITF[34,42], western blotting using a pan-anti-acetyl lysine antibody of immunoprecipitated GFP-tagged MITF co-expressed with either the CBP, p300 or GCN5 acetyl transferases revealed acetylation of MITF, but not GFP, when MITF was co-expressed with CBP or p300 (Fig. 1a). No significant acetylation was detected when GCN5 was co-expressed. Mass spectrometry analysis of immunoprecipitated MITF revealed acetylation of four residues, K33, K91, K206 and K243 (Fig. 1b, Supplementary Fig. S1), but no other modification (e.g., methylation) on these sites was detected in this or other experiments.

Of the acetylated residues identified, only K206 lies within the basic region that determines MITF sequence-specific DNA-binding activity. This residue is highly conserved between species and in all members of the MITF subfamily of bHLH-LZ transcription factors including TFEB and TFE3 (Fig. 1c), but not in other bHLH-LZ proteins such as USF1, MYC, or MAX, suggesting it may play a distinct regulatory role in the MITF-subfamily. Moreover, a DNA-MITF DNA-binding domain (DBD) co-crystal structure indicates K206 contacts the phosphate backbone within the basic region[41] (Fig. 1d). Significantly, a

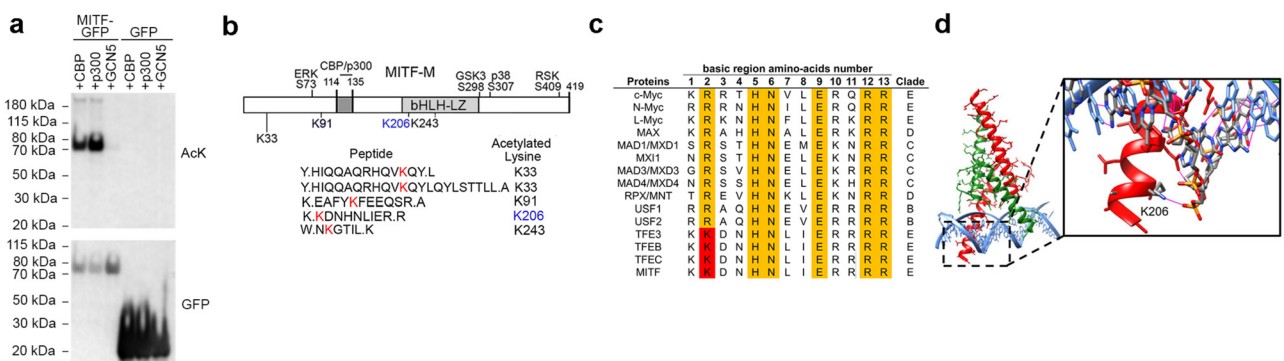

**Fig. 1 | Acetylation of MITF at K206. a** Western blot using anti-acetyl lysine or anti-GFP antibodies of immunoprecipitated GFP-tagged MITF co-expressed with CBP, p300 or GCN5 in Phoenix-AMPHO cells as indicated. AcK; Acetyl-Lysine. This experiment was repeated 3 times with similar results. Source data for Western blots are provided in Supplementary Fig. S6. **b** Schematic showing locations of MITF acetylation sites (red) detected in this study, and AcK206 in blue. Acetylated peptides derived from Mass Spec analysis are indicated below and phosphorylation sites and the p300 binding motif are indicated above. **c** Alignment of amino acids in the basic region of related bHLH-LZ family members. Yellow highlights highly conserved residues, and the red highlight (corresponding to K206 in MITF) indicates lysines conserved only in the MiT subfamily. **d** Depiction of the MITF-DBD-DNA co-crystal structure highlighting the K206-phosphate backbone interaction. MITF monomers are depicted in red or green, and DNA in blue.

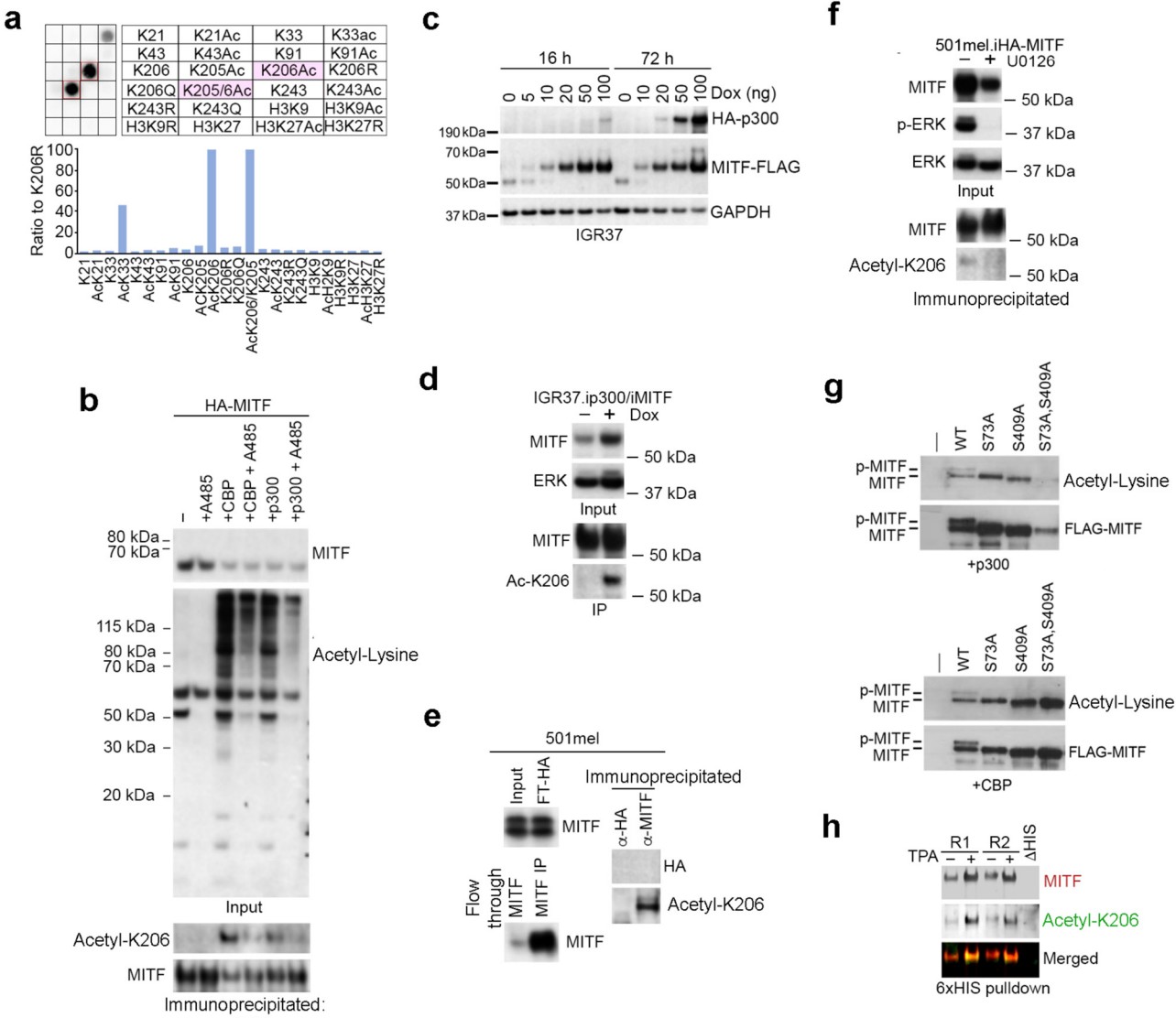

**Fig. 2 | Confirmation of MITF acetylation at K206. a** Peptide array of 14 amino acid peptides containing indicated acetylated or non-acetylated residues probed with anti-acetyl K206 antibody with AcK206 in pink boxes. Signal was quantified based on the chemiluminescence signal. **b–h** Western blots probed with indicated antibodies. **b** HA-MITF expressed in Phoenix-AMPHO cells alone or co-expressed with CBP or p300 +/− 20 nM A485 probed with anti-acetyl lysine or anti-HA anti-bodies before (top panels) or after (bottom panels) immunoprecipitation using anti-HA. Experiment performed once as presented, but repeated with similar results by using A485 to block TPA-induced MITF acetylation. **c** IGR37 cells expressing doxycycline-inducible FLAG-tagged MITF and HA-tagged p300 induced for 16 h or 72 h using indicated amounts of doxycycline. Experiment repeated twice with similar results. **d** Immunoprecipitation of HA-tagged MITF from IGR37 cells engineered to express doxycycline inducible p300 and HA-MITF. Experiment performed once. **e** Endogenous MITF immunoprecipitated from 501mel cells using anti-MITF antibody or anti-HA as an isotype antibody control and probed with

indicated antibodies. Input is shown to the left. FT indicates 'flow-through'; IP indicates immunoprecipitated. Experiment performed once. **f** Immunoprecipitation of HA-tagged MITF from 501mel cells treated or not with 10 μM U0126 as indicated. Inputs and immunoprecipitated protein were western blotted using the indicated antibodies with anti-ERK used as a loading control. Experiment has been repeated twice with similar results. **g** Expression of FLAG-tagged WT or mutant MITF co-expressed with p300 (left) or CBP (right) in Phoenix-AMPHO cells and probed with anti-acetyl lysine or anti-MITF antibody. Experiment has been performed once as presented, but repeated using S73A and S73A, S69A mutants with similar results. **h** HIS-tagged MITF from 501mel cells engineered to express ectopically inducible 6xHIS tagged MITF treated with or without 200 nM TPA for 6 h purified using nickel beads from urea-dissolved cells and probed with anti-acetyl K206 or anti-MITF antibodies. Two replicates are shown, and purification of a non-his-tagged MITF (ΔHis) was used as a negative control. Source data for Western blots are provided in Supplementary Fig. S6.

heterozygous lysine to glutamine substitution (K206Q), that is widely accepted as an acetylation mimetic, has been identified in a family with Waardenburg syndrome Type 2[43], characterized by pigmentation abnormalities and hearing loss owing to defective melanocytes in the inner ear. Collectively these observations suggest that acetylation of K206 could potentially play a key role in modulating MITF function.

To validate the mass spectrometry results we generated an anti-acetyl K206 antibody and tested its activity against a range of MITF peptides. The results revealed that the antibody was highly selective,

strongly recognizing peptides containing acetyl-K206, and recognizing a peptide containing acetyl K33 at least 2.5-fold less well (Fig. 2a). No activity was detected against non-acetyl K206, nor against a range of other acetyl or non-acetyl control peptides derived from MITF or histone H3. To confirm acetylation of MITF at K206, we immunoprecipitated HA-MITF co-expressed with CBP or p300, with or without the selective p300/CBP inhibitor A485. The results revealed that in the input control, p300 or CBP increased global acetylation detected by a pan-acetyl-lysine antibody that was diminished by A485 (Fig. 2b, upper

panels). After immunoprecipitation of MITF using an anti-HA antibody, probing with the anti-acetyl MITF K206 antibody revealed MITF acetylation at this site was decreased in the presence of A485 (Fig. 2b, lower panel). Immunoprecipitation of MITF from an IGR37 human melanoma cell line engineered to express inducible p300 and HA-MITF (Fig. 2c) so as to partially maintain their relative expression levels revealed an increase in acetyl K206 MITF after induction using doxycycline (Fig. 2d), confirming a role for p300 in MITF acetylation. We were also able to detect acetylation of endogenous MITF after immunoprecipitation using anti-MITF antibody and an anti-HA antibody as an isotype control (Fig. 2e). While anti-HA failed to immunoprecipitate endogenous MITF as expected, the anti-MITF antibody pulled down MITF (bottom left panel) that when probed with the anti-acetyl K206 antibody (right panel) revealed that MITF is acetylated at this residue. Since p300 acetyl transferase activity is activated by MAPK signaling[29,35] downstream from BRAF or NRAS, we also repeated the immunoprecipitation experiment using HA-MITF from cells treated or not with the MEK inhibitor U0126. The results indicated that MITF acetylation at K206 was reduced when the MAPK pathway was inhibited (Fig. 2f). This was unlikely to arise because of previously described phosphorylation of MITF by MAPK signaling at either S73 or S409[34,44], since mutation of either site to alanine alone or in combination failed to block MITF acetylation by p300 or CBP detected using anti-acetyl lysine antibody (Fig. 2g). Finally, we repeated the experiment using a 501mel cell line stably expressing HIS-tagged MITF at endogenous levels[29]. This enables the direct purification of MITF on nickel beads after solubilization of cellular proteins using 6 M guanidinium HCl, and circumvents problems associated with the previously observed poor extractability of endogenous MITF from nuclei[29]. In this experiment, 200 nM TPA, previously shown to promote melanogenesis[45], was used to increase MITF expression. The result (Fig. 2h) confirmed that MITF K206 is acetylated in the absence of ectopic p300 or CBP.

## Single molecule tracking reveals K206 regulates MITF chromatin association in vivo

MAPK-dependent acetylation of MITF at K206 is likely to have functional consequences, in part because of the pigmentation defects in human K206Q heterozygotes. Because K206 makes a phosphate backbone contact in the DNA co-crystal structure, we asked whether a K206R non-acetylatable mutation, or the K206Q mutation that mimics constitutive acetylation, might affect the ability of MITF to bind DNA. To do this we initially used a single molecule tracking (SMT) assay[46] to examine MITF WT and K206 mutant dynamics and their chromatin association in live cells. By fusing MITF to the HALO-tag labeled at sub-saturating concentrations with JF549, a bright photostable fluorescent ligand[47], we can estimate the duration of binding events, the fraction of immobilized MITF bound to chromatin, and the diffusion properties of the protein[48–50]. We therefore established a 501mel human melanoma cell line stably expressing doxycycline-inducible HALO-tagged MITF WT and K206 mutants (Fig. 3a). We also included unconjugated HALO-tag as an additional control, and in some experiments also used a non-DNA-binding MITF mutant (Δbasic) lacking the basic region that makes direct DNA contacts so that we could determine the relative contribution of direct DNA-binding versus non-DNA associations mediated, for example, by protein-protein interactions. In all constructs we also included the SV40 T-antigen nuclear localization signal[51] since perturbation of the MITF basic region can affect nuclear import[52]. We then tuned the doxycycline concentration to induce expression of the HALO-tagged MITF WT and K206 mutants at comparable levels (Fig. 3b) and imaged single MITF molecules under two different regimes. First, to evaluate how K206 acetylation affects the mobility of the MITF we acquired movies at 100 fps, using a 5 ms laser exposure. Single molecule tracks were analyzed in terms of the distributions of displacements (Fig. 3c), which were fit with a three component diffusion model as previously described[53–55], where the slowest component

describes immobilized molecules (on chromatin or other nuclear structures), and the two others represent diffusing MITF, possibly slowed down by transient interactions (faster than the acquisition framerate). WT MITF (Fig. 3c, top left) displayed a much larger fraction of immobilized molecules than unconjugated HALO-tag (Fig. 3c, bottom right; median 44% and 13% respectively) with a median diffusion coefficient $D_{bound} = 0.04\ \mu m^2/s$, comparable to that previously measured for tightly chromatin-bound proteins, such as histones. 34% of WT MITF was found in a slow diffusion state with a median $D_{slow} = 0.46\ \mu m^2/s$, and the remaining molecules were in a fast diffusion state with a median $D_{fast} = 3.4\ \mu m^2/s$ (Fig. 3c, d). Compared to WT MITF, the K206R mutant (Fig. 3c, bottom left) displayed a non-significant decrease in its bound fraction (Fig. 3d), and similar diffusion coefficients (Fig. 3d). The MITF K206Q mutant, instead exhibited a significantly decreased bound fraction (Fig. 3d). Notably, repeating the same experiment using inducibly expressed HALO-MITF WT and mutants (Supplementary Fig. S2) in a cell line expressing very low endogenous levels of MITF (IGR39)[20] recapitulated the results obtained in the MITF^High 501mel cells (Supplementary Fig. S3a, b). This included a drop in the bound fraction observed for the MITF K206Q mutant, compared to WT, this time also accompanied by an increase in the MITF diffusion coefficients. Together, these results highlight that the MITF K206Q mutant is generally more mobile than MITF-WT in the nucleus of living cells.

To better characterize MITF binding kinetics, we next collected single molecule movies at 2 fps, using a laser exposure of 200 ms to blur out the diffusing molecules[56] and to quantify the dwell times of immobilized molecules, including the non-DNA-binding Δbasic mutant (Fig. 3a) as an additional control. Since the measurements taken over a long period can be affected by photobleaching, we used a histone H2B-HaloTag as an further control since a large quota of H2B is stably incorporated into chromatin, and then corrected the MITF residence times using the slowest decay rate observed for H2B-HaloTag, as recently described[57]. The results obtained in the 501mel cell line are shown in Fig. 3e, with quantification of binding events lasting more than 100 s shown in Fig. 3f (left panel), and the average survival time of bound molecules presented in Fig. 3f (right panel). These data enable us to generate several conclusions. First, based on the results corrected for photobleaching, over 40% of bound WT MITF exhibits a long residency time in excess of 100 s, with an average survival time of 46 s. Second, such long-lived binding was only modestly decreased for the K206R mutant (average survival time 42 s, 40% of molecules exceeding 100 s of residence time), but was decreased even further for the K206Q mutant (average survival time 36 s, 33% of molecules exceeding 100 s of residence time). Third, the Δbasic DNA-binding MITF mutant displays more transient immobilization events than WT MITF and MITF-K206 mutants, although a proportion (about 9%) of the protein still exhibits a residence time of greater than 100 s. This indicates that the majority of the stably bound MITF is likely interacting with DNA via the basic region, but a contribution of long-lived residence time is made by residues outside the basic region that presumably participate in protein-protein interactions. Of note, another bHLH-LZ transcription factor, USF1, as well as an unrelated transcription factor, p53, display a much faster dissociation from chromatin than WT MITF, indicating that tight binding to DNA in vivo might be a peculiar property of MITF. This is consistent with previous observations[29] indicating that unlike USF1, MITF is poorly extractable from cell nuclei even under very harsh conditions.

Repeating the residence time measurement in the MITF-low IGR39 cell line (Supplementary Fig. S3c,d), largely recapitulated these observations. In this setting, WT MITF displays a long mean survival time of approximately 45 s, similar to that of the K206R mutant and to that obtained with WT MITF in the MITF^High 501mel cell line. The mean survival time is again significantly reduced to around 33 s for the acetylation mimetic MITF K206Q mutant (Supplementary Fig. S3c,d). Thus, the K206Q mutant spends a shorter time at its binding sites,

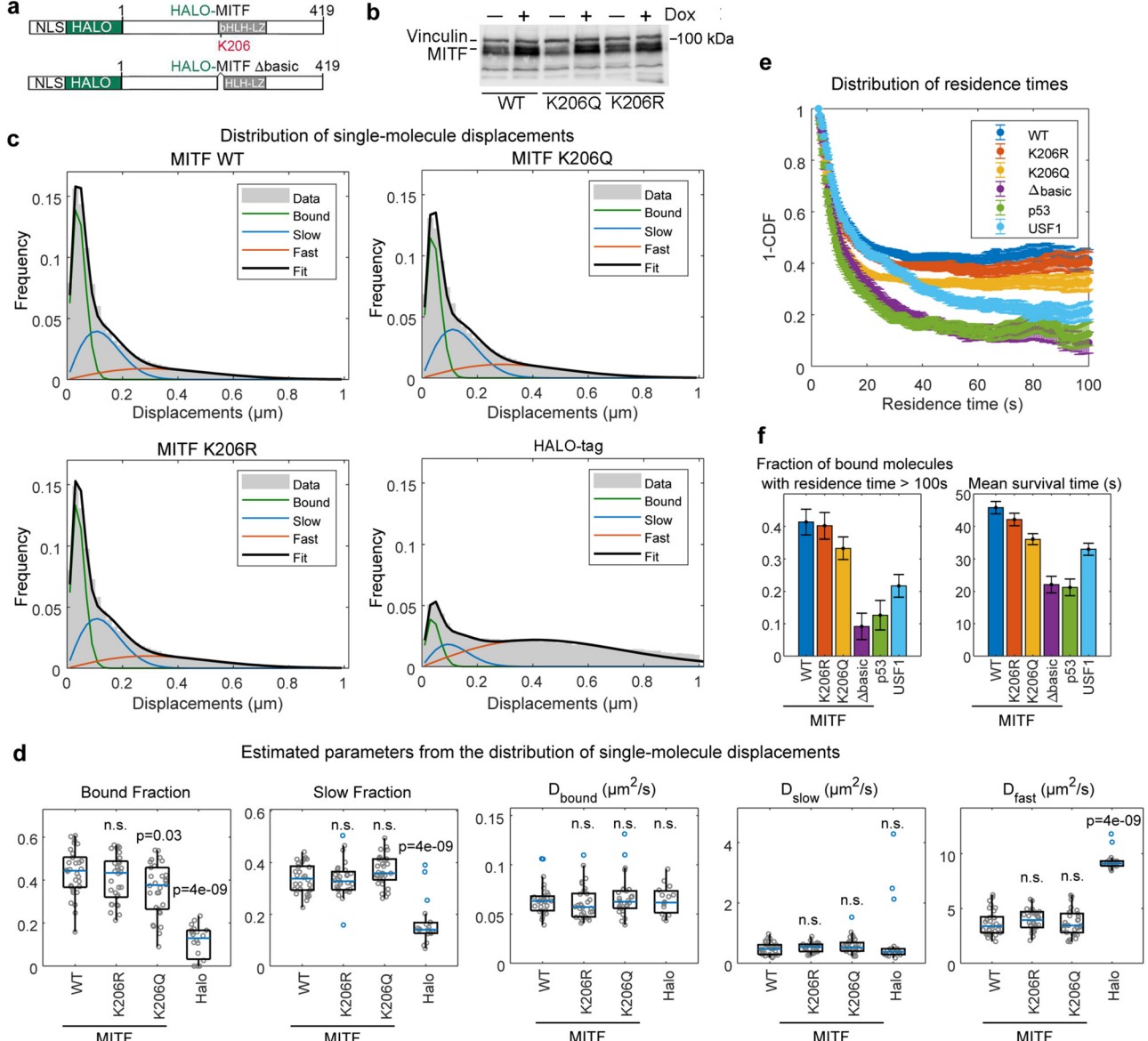

**Fig. 3 | Live-cell single-molecule tracking of HALO-tagged MITF. a** HALO-tagged MITF expression vectors. NLS; nuclear localization sequence. **b** Western blot of HALO-tagged MITF WT and mutants in 501mel cells with or without treatment with Doxycycline used to ensure similar expression levels performed in parallel with the SMT analysis shown in this figure. Vinculin was a loading control. Induction of HALO-MITF by Dox in 501mel cells has been performed independently at least 3 times with similar results. For Source Data see Supplementary Fig. S6. **c** Single molecule tracking movies collected at 100 fps tracks were extracted using the FiJi/ImageJ plugin TrackMate and analysed in terms of the distribution of single-molecule displacements between consecutive frames that was then fit with a three component model (one immobile component and two diffusing components), to generate quantitative estimates for HaloTag, WT MITF and mutants. **d** Quantitative estimates derived from SMT using WT and K206 HALO-tagged MITF for the fraction of molecules in the bound and slow states and the respective diffusion coefficients. Each point represents a single cell, the blue line the median, the box limits represent upper and lower quartiles, and whiskers extend between Q1 − 1.5 IQR and Q3 + 1.5 IQR, where IQR is the interquartile range. For HaloTag, WT MITF, K206R and K206Q mutants respectively $N_{replicates} = 2$; $N_{cells} = 19$; 30; 30; 30; $N_{jumps}$: 130387; 245200; 210929; 172286. Statistical test: non-parametric Kruskal-Wallis (two-sided). **e** Slower movies (frame rate 2 fps, laser exposure 200 ms) were acquired to calculate the distribution of residence times for immobile MITF molecules, following photobleaching correction using data collected on H2B-HaloTag (see Methods). **f** The fraction of bound molecules displaying a residence time longer than 100 s (left) and the (restricted) mean survival time (right). $N_{replicates} = 2$; $N_{cells} = 30$, 30, 30, 30, 35, and 36, and $N_{molecules} = 3836$; 3429; 3219; 1185; 3809; 1225 for WT MITF, K206R, K206Q, Δbasic, USF1 and p53 respectively; error bars (SEM) were calculated by the jacknife approach (see Methods); each SMT experiment was performed at least twice after adjustment of doxycycline concentration to ensure similar MITF expression levels.

while the K206R mutant displays nuclear dynamics more similar to WT MITF.

### Acetylation of MITF K206 differentiates between M-box and CLEAR-box motifs

To determine the effect of acetylation of K206 on MITF DNA-binding directly we bacterially expressed the MITF DNA-binding and dimerization domain (DBD) and used the purified protein in in vitro DNA-binding assays. A modified bacterial expression system, in which introduction of an amber stop codon into the K206 position enabled acetylation of K206 to be genetically encoded, allowed us to express and purify the MITF DBD specifically acetylated to high efficiency at K206 (Supplementary Fig. S4). We also expressed and purified the wild-type (WT) MITF DBD, a non-acetylatable K206R mutant, and the

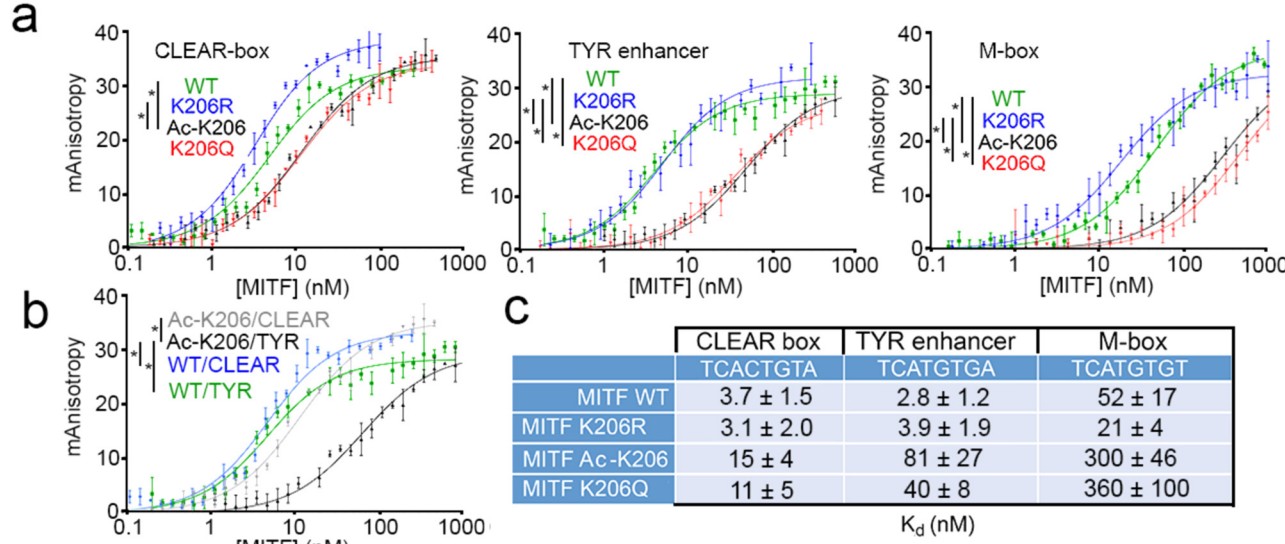

**Fig. 4 | Acetylation of K206 specifically decreases binding to an M-box.**
**a**, **b** DNA-binding affinity of bacterially expressed and purified MITF WT and mutant DNA-binding domains determined using fluorescence anisotropy. Representative titration curves of each fluorescein-labeled oligonucleotide with MITF WT and mutants. The reported anisotropy values are the average of triplicate measurements, from which the base line corresponding to the anisotropy of the free fluorescent probe was subtracted. (*) $p < 0.05$ based on an unpaired two-tailed t-test with error bars indicating SD calculated for each measurement point. **a** Graphs are presented in order to optimally compare the affinity of each MITF variant for three different DNA motifs investigated here. **b** Curves represented on this graph correspond to the titration of two DNA motifs by MITF WT and acetylated MITF. **c** Quantitative determination of DNA-binding affinities of MITF variants, measured as $K_D$ (nM). The $K_D$ values correspond to the means of three independent measurements and the +/− error numbers represent the standard deviations. Source data are provided in a Supplementary Source Data file.

Waardenburg-associated K206Q mutant that was expected to mimic acetylation by breaking the phosphate backbone contact. These were then used in fluorescence anisotropy assays together with three fluorescently labeled oligonucleotides. These contained the full palindromic 8 bp TCACGTGA binding motif termed the CLEAR-box that is associated with genes other than those implicated in pigmentation; a related TCATGTGA element associated with the melanocyte-specific *TYROSINASE* (*TYR*) enhancer that retained the flanking 5′T and 3′A bases important for MITF binding[58]; and the TCATGTGT M-box sequence[59] found in differentiation-associated promoters. Increasing the amounts of the purified WT, K206R and K206Q mutants and Ac-K206 proteins allowed us to determine the relative affinities for each type of element. Examples of the binding curves are shown in Fig. 4a, b and a summary of the relative affinities derived is presented in Fig. 4c. The bacterially expressed and therefore non-acetylated WT MITF exhibited only a moderate reduction in affinity between the CLEAR-box and *TYR* enhancer, but has over a 10-fold reduced affinity for the promoter M-box motif. Similar results were obtained using the non-acetylatable K206R mutant. By contrast, Ac-K206 MITF clearly distinguished between the two sequence elements: binding to the TCACGTGA CLEAR-box was reduced around 3-fold compared to the non-acetylated WT protein, but was diminished a further 5-fold using the *TYR* TCATGTGA motif while binding to the M-box was 20-fold reduced compared to the CLEAR box. The effect of acetylation on MITF DNA binding was largely reproduced using the K206Q mutant. These results revealed that while acetylation of MITF K206Q moderately reduces binding to the CLEAR box, binding to the M-box present in differentiation-associated promoters was much more severely affected. Thus, modification of K206 can potentially enable MITF to discriminate between these two classes of regulatory element. The reduced DNA binding affinity of the K206Q mutant, reflected in its reduced residence time determined using SMT analysis, may therefore account for the Waardenburg disease associated with this variant.

### Modification of MITF K206 affects melanocyte development
Since Ac-K206 MITF exhibits reduced DNA-binding affinity we anticipated that the Waardenburg-associated K206Q mutation would exhibit defects in melanocyte development. To test this, we used a zebrafish assay in which MITF WT or mutants are used to complement the absence of MITF in an *mitfa*-null *nacre* zebrafish that lacks all neural crest-derived melanocytes[60]. To this end, WT or K206 mutant (K201 in fish) zebrafish MITF were transiently expressed from the fish *mitfa* promoter (Fig. 5a) and the effects on melanocyte number in the larval fish assessed 5 days post-fertilization. The results (Fig. 5b, c) revealed that whereas WT and K201R mutant MITF complemented the absence of endogenous MITF and generated a normal pattern of melanocytes, the K201Q mutant failed efficiently to complement the absence of MITF, consistent with it possessing a defective capacity to regulate gene expression.

### K206 modification regulates genome-wide distribution of MITF
The results so far suggest that acetylation of MITF would lead to reduced DNA binding overall, but that binding to differentiation-associated genes would be significantly more affected. Since binding of the K206Q mutant accurately mimicked binding of the acetylated K206 protein in vitro, we next established human 501mel melanoma cells in which HA-tagged WT MITF, as well as the K206Q and K206R mutants, were stably expressed from a doxycycline-inducible promoter. By using an inducible promoter, we were able to carefully titrate the levels of MITF WT and mutants expressed (Fig. 6a). Since the ectopic HA-epitope tagged MITF migrates slower when analyzed by SDS PAGE, we determined that 20 ng doxycycline induced a similar level of ectopic WT and mutant MITF to the endogenous protein present in uninduced 501mel cells (Supplementary Fig. S5a). Using these inducible MITF cell lines we performed duplicate ChIP-seq experiments at 0, 20, and 100 ng doxycycline for the WT and each mutant, with the replicate results obtained showing high concordance (Supplementary Fig. S5b).

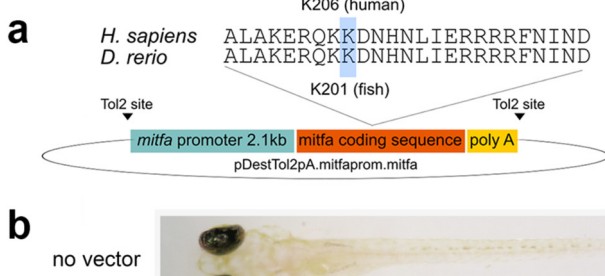

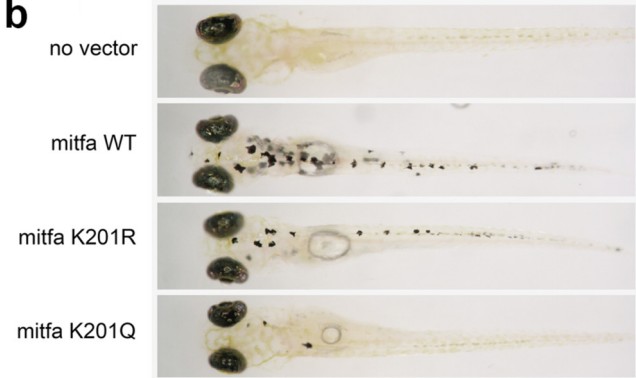

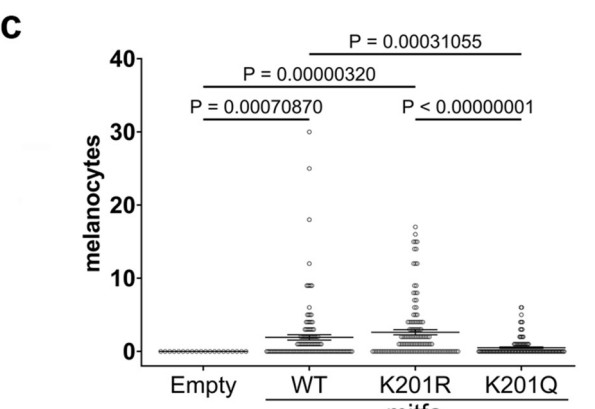

**Fig. 5 | K206 affects melanocyte development. a** Amino acid sequence alignment of human and zebrafish MITF showing K206 (human) and K201 (fish) conservation. The fish *mitfa* coding sequence was placed under the control of the fish *mitfa* promoter. **b** Complementation of neural crest *mitfa*-null *nacre* zebrafish using MITF WT and K201 (K206 in human MITF) mutants. **c** Quantification of numbers of melanocytes in the zebrafish complementation assay using Jitter plots in which the dots in the plots represent the numbers of melanocytes in each rescued embryo with at least 1 melanocyte, the line shows mean +/−SEM. Statistical test: Kruskal-Wallis test and Dunn's multiple comparisons test. Data were obtained from a total of 7 experiments in which a total of 411 fish were analysed: 20 for empty vector control; 132 for WT-MITF; 201 for MITF K201R; and 127 for MITF K201Q. Source data are provided in the Supplementary Source Data file.

As expected, the read density of the WT MITF ChIP showed that increasing the levels of doxycycline led to increased MITF DNA binding reflected in an increasing peak score (Fig. 6b, left panel), an increase in total numbers of peaks called, and corresponding increase in concordance between the replicates (Supplementary Fig. S5c). This was reflected in binding at individual genes known to be MITF targets (Fig. 6c, left panels; Supplementary Fig. S5d, left panels). For example, at 0 ng doxycycline, when the HA-tagged MITF is undetectable by western blotting (Fig. 6a), a highly specific ChIP signal is observed at the *BLOC1S* and *MLANA* genes that contain TCACGTGA or BCACGTGA MITF binding sites respectively, but a peak was not readily discernable at *DCT* with a TCATGTGC site. As the level of MITF is increased using 20 ng or 100 ng doxycycline the peak score increases at *BLOC1S* and at *MLANA*, and at *DCT* specific binding now becomes apparent. As shown

for other transcription factors[2], most likely nucleosome positioning or cooperating co-factors make significant gene-specific contribution to MITF binding affinity at any locus. Nevertheless, the results indicate that increasing MITF levels, for example as observed during melanocyte differentiation in response to cAMP signaling downstream from melanocortin 1 receptor signaling[61,62], allows binding to differentiation-associated genes such as *DCT* that is not detected at lower levels of MITF.

We next compared the binding of WT MITF to that of the non-acetylatable K206R mutant and the Waardenburg-associated K206Q acetylation mimetic. Although with lower peak height than WT MITF, binding by the K206R mutant was detected at the lowest concentration of MITF to genes such as *BLOC1S* or *MLANA*, though not at *DCT* (Fig. 6c, middle panels). At 20 ng doxycycline, when ectopic MITF was expressed to levels similar to the endogenous protein, at many genes K206R bound better than WT (Fig. 6c, d; Supplementary Fig. S5d), consistent with a proportion of WT MITF being acetylated (Figs. 1 and 2). By contrast, the K206Q mutant exhibited a decrease in global DNA binding, consistent with a reduced residence time in cells (Fig. 3e,f) and lower DNA binding affinity in vitro (Fig. 4). While binding was detected by the K206Q mutant at *BLOC1S* and *MLANA*, with a peak height around 30% of WT at 20 ng doxycycline, binding to the differentiation-associated genes *DCT* (Fig. 6c, lower panel) or *PMEL* (Supplementary Fig. S5d, lower right panel) was barely detected above background at any concentration of doxycycline. Given that at 20 ng doxycycline binding by WT MITF was intermediate between that of the non-acetylatable K243R mutant and the acetylation mimetic K243Q mutant, our results are consistent with a significant proportion of MITF being acetylated in 501mel cells, as observed using the anti-acetyl K206 antibody (Fig. 2e).

The DNA-binding affinity measurements (Fig. 4) indicated that acetyl K206 MITF or the K206Q mutant can distinguish between a CACGTG motif and the lower affinity CATGTG E-box. To determine whether the K206Q and K206R mutants differentially recognized these motifs in cells, we identified the sequences beneath the peaks after induction using 0, 20, or 100 ng doxycycline. The results for the WT protein indicated that as MITF levels were increased, there was an increase in the proportion of lower affinity binding sites recognized, with a shift away from the canonical TCACGTGA motifs towards TCACGTGB or TCATGTGA elements (Fig. 6e). As expected the K206R mutant behaved similarly to the WT protein, although at 0 ng doxycycline there was a greater proportion of TCACGTGA motifs bound. By contrast, although a largely similar shift was observed on increasing the levels of the K206Q mutant, the vast majority of sites recognized were TCACGTGA. These observations were confirmed when analyzing the global consensus beneath the WT and mutant MITF peaks at 20 ng doxycycline that revealed that the consensus for the WT and K206R mutants reflected a mix of CACGTG and CATGTG core E-box motifs, while that of the K206Q mutant was restricted to the high affinity CACGTG element (Fig. 6f). A similar result was obtained by analysis of peak score versus motif incidence that revealed that the WT and K206R mutants behaved similarly, though with the K206R mutant exhibiting somewhat better binding than WT MITF (Fig. 6g). Again, the K206Q mutant bound fewer sites and those that were bound were primarily CACGTG motifs.

## Discussion

Transcription factors play a critical role in establishing and maintaining specific cellular phenotypes. To do so they must identify specific sets of binding sites in genes that determine cell identity. In melanoma, MITF has been implicated in regulating many biological processes including reprogramming metabolism, promoting survival, proliferation, differentiation and autophagy, and suppressing invasion and senescence[4]. The current rheostat model of MITF function suggests that different levels of MITF expression or activity enables cells to

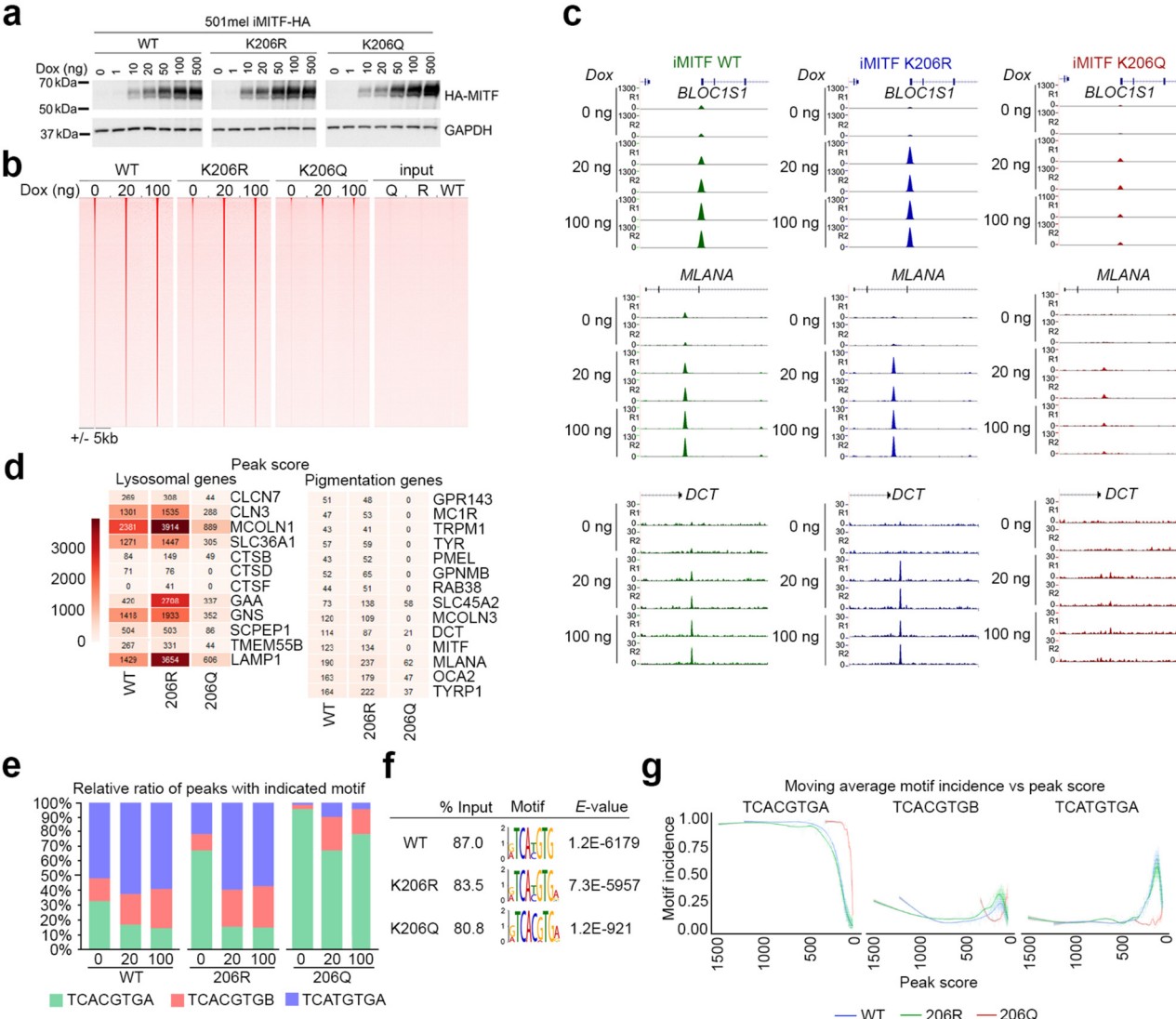

**Fig. 6 | K206 controls MITF genome-wide distribution. a** Western blot of 501mel melanoma cell lines stably expressing HA-tagged doxycycline-inducible MITF WT and K206 mutants after induction with indicated concentrations of doxycycline. MITF inducibility by Dox is highly reproducible and has been repeated independently at least 5 times. Source data are provided in Supplementary Fig. S6. **b** Heatmap visualization of MITF WT and K206 mutants read density derived from HA-MITF ChIP-seq using indicated concentrations of doxycycline. Input controls are shown to the right. **c** UCSC genome-browser screenshots of ChIP-seq profiles of HA-MITF WT and mutants at different concentrations of doxycycline as indicated.

R1 and R2 indicate two independent replicates. **d** ChIP-seq peak scores for a set of lysosomal or pigmentation genes as indicated for MITF WT and mutants induced using 20 ng doxycycline. **e** Relative ratio of ChIP peaks with indicated motifs for MITF WT and mutants expressed at 0, 20, and 100 ng doxycycline. **f** Consensus motifs detected beneath the ChIP peaks for MITF WT and mutants. **g** Scatter plots comparing ChIP-seq peak score versus moving average of relative motif incidence (100 peaks window) for MITF WT and mutants. To aid visual comparison the data points were fitted with smoothed line using generalized additive models. The 95% confidence interval is shown in gray shade around the fitted smoothed line.

adopt distinct phenotypes[10]. This model is supported by the observation that a reduction in MITF function using a temperature sensitive mutant in zebrafish causes differentiated melanocytes to re-enter the cell cycle[60,63] and that MITF-independent states characterize persister states in residual disease that are similar to those generated in response to BRAFi[64]. Our results suggest that the CATGTG M-box motifs associated with the promoters of differentiation genes are lower affinity binding sites, whereas CACGTG E-boxes are higher affinity and can therefore be recognized at lower protein levels. However, in addition to MITF levels, the ability of MITF to control its target genes will also be regulated by post-translational modifications.

Significantly we find that the ability of MITF to bind DNA is inhibited by CBP/p300-mediated acetylation of K206. This residue lies within the basic region that mediates both base-specific contacts to provide sequence specific binding, as well as phosphate backbone interactions that increase DNA binding affinity without directly

affecting base recognition. However, while acetylation of K206, that makes a phosphate backbone contact, moderately reduces the affinity to a CACGTG E-box or TCACGTGA enhancer element, it drastically reduces binding to CATGTG M-box motifs. The reduction in Ac-K206 DNA binding in vitro was confirmed in cells using a K206Q mutant and SMT analysis irrespective of whether endogenous MITF is present. Notably, the in vitro binding results were reflected in vivo using ChIP-seq where recognition of CATGTG motifs by the K206Q mutant was largely blocked while binding to genes containing CACGTG elements was retained, though with a reduced peak height. The ability of K206 acetylation to differentiate between CATGTG and CACGTG elements suggests that regulation of MITF acetylation may be instrumental in determining MITF's repertoire of target genes. Consistent with activation of BRAF or NRAS upstream from ERK promoting melanocyte proliferation and de-differentiation, MITF interaction with CBP/p300 is promoted by ERK-mediated phosphorylation of MITF[34] and CBP/p300

acetyl transferase activity is stimulated by MAPK signaling[29,35]. However, the activity of CBP/p300 is also inhibited by deacetylation mediated by SIRT1[65], an MITF-activated target gene[66], suggesting that regulation of MITF acetylation by CBP/p300 may be subject to an autoregulatory feedback loop.

Our results also suggest that MITF may represent a distinct class of transcription factor that is tightly bound within the nucleus through both DNA and non-DNA interactions with over 40% of molecules exhibiting a residence time of greater than 100 s. This is remarkably long for a transcription factor where SMT has determined typical residence times of few seconds[67]. The known exceptions are the serum response factor SRF after serum stimulation in fibroblasts ($\tau_{res}$ 1–4 min)[68], polycomb repressive complex 1 (PRC1) ($\tau_{res} > 100$ s)[69] and CTCF ($\tau_{res}$ around 60–120 s)[70,71]. The long residency time of MITF is not related to the structure of the MITF DNA binding and dimerization domain nor the E-box motifs recognized, since USF1, which also contains a bHLH-LZ motif and can bind the same sites as MITF[72], is readily extracted from nuclei and binds with a residence time distribution similar to that of the K206Q MITF mutant. It seems likely therefore, that a significant contribution to MITF's residency time is made by protein–protein interactions. Given the role of CTCF in determining and stabilizing chromatin topology though binding to boundary elements[71,73], and the role of PRC1 in 3D chromatin organization and control of cell identity[74–77], it is plausible that MITF's tight association with nuclear structures[29] and long residency times reflects a similar role in establishing chromatin conformations that determine the melanocyte lineage. This would be consistent with observations made on the residence time of the neuronal lineage-determining factor Ascl1 using in vivo competition assays[78]. These authors suggested that the long residence time of Ascl1 in non-dividing cells, that was in the order of hours, might be important for securing the differentiated state of cells, while in proliferating cells residence times would be reduced to generate flexibility in the transcriptional response. In this respect we note acetylation by p300/CBP is controlled by pro-proliferative MAPK signaling[29,35]. Consequently, in non-dividing differentiated cells where MAPK signaling will be reduced, it seems possible that MITF's long residence time will be further increased. Moreover, if endogenous MITF has a very long residence time, its ability to be displaced by the expression of ectopic MITF would be reduced. If so, this might explain the observations[79,80] that while depletion of MITF leads to a reduction in expression of its target genes, ectopic MITF fails to regulate the same genes.

Finally, in addition to deciphering how K206 acetylation affects MITF target specificity, our results may also be relevant for understanding the genetics of Waardenburg's disease. In this respect, it is clear that the Waardenburg's-associated K206Q mutation[43], by mimicking K206 acetylation leads to reduced MITF DNA residence time and consequently to a diminished capacity to promote the biological functions associated with MITF leading to the developmental defects associated with the disease.

In summary, our results highlight a mechanism by which post-translational modification of MITF regulates its ability to bind its targets and thereby control its ability to coordinate the gene expression programs that underpin melanocyte and melanoma proliferation and differentiation.

## Methods

### Plasmids
pCS2-6xMyc-MITF used for mass spectrometry has been described previously[81]. pCMV5-3xHA-Mitf was PCR cloned from Mitf cDNA using EcoRI (NEB; Cat#R3101L). pEGFP-Mitf was generated via subcloning from pCMV5-3xHA-Mitf using EcoRI into pEGFP-N1 (Clontech). pETM-11-MITFΔN180ΔC296 has been described[41], as has the doxycycline inducible PiggyBac 3xHA-Mitf expression vector used for ChIP-seq[17], and the 3xHA-MITF-3xFLAG-6xHIS used for acetylation analysis and

the Halo-tagged WT MITF[29]. The K206 mutants were derived by site-directed mutagenesis (Agilent #210519). pcDNA3-3 × HA-p300 was a gift from Bernhard Luescher. All plasmids have been sequenced and verified prior to use.

### Cell lines
HEK293 cells used for the MITF mass spectrometry analysis were obtained from Pierre Chambon's laboratory. 501mel melanoma cells were obtained directly from Ruth Halaban (Yale) who's lab originated this cell line. 501mel cells and their derivative cell lines were maintained in RPMI-1640 (Gibco#61870-010), 10% fetal bovine serum (FBS, Biosera) at 37 °C in humidified incubator with 10% CO2. Reverse-transfection was perform using 3:1 ratio of FUGENE VI (Promega # E2692):DNA suspended in Opti-MEM (GIBCO #31985-062) according to the supplier's protocol. Stable inducible cell lines were generated from transiently-transfected cells using Geneticin (1 mg/ml) (GIBCO #10131-027), Puromycin (5 μg/ml) (GIBCO #A11138-02).

All cell lines were tested monthly for mycoplasma and were negative and authentication performed using short tandem repeat analysis.

### Antibodies
Primary antibodies used were: α-Acetylated-Lysine (RRID: AB_331805); MITF (RRID: AB_570596); HA (RRID: AB_514506); FLAG (RRID:AB_262044); GAPDH (RRID: AB_627679); VINCULIN (RRID: AB_10976821) and GFP (RRID:AB_303395). MITF-K206Ac antibody was raised in-house. Secondary HRP antibodies were procured from Biorad (#170-6516 & #170-6515) and Secondary Alexa dye-labeled antibodies procured from Invitrogen (#A21202 & A10040).

**Zebrafish.** Plasmid DNA (62.5 ng/μl) comprising the zebrafish promoter driving the fish MITF cDNA was mixed with Tol2 mRNA (75 ng/ μl). 2 nl of the mixture was injected into 1-cell stage *mitf*-null *nacre* embryos. Injected embryos were grown at 28 °C for 5 days. On day 5, embryos were briefly exposed to white light to contract melanocytes and were then imaged before being fixed in 4% PFA. The total number of surface melanocytes in the head, trunk, and yolk sac was counted. Three independent experiments were performed. All zebrafish experiments are performed in accordance with the *Animals (Scientific Procedures) Act* 1986, and approved by the University of Edinburgh Animal Welfare and Ethical Review Body. Zebrafish AB/TPL lines were bred, raised and maintained as described[82].

**Western blotting.** Whole cell extract (WCE) were prepared from trypsinised cells, and washed and centrifuged cell pellets dissolved in 4× Laemmli sample buffer or 4× NuPAGE® LDS Sample Buffer (Novex® #B0008) + 20% β-mercaptoethanol, added fresh before use. Extracts were boiled at 95 °C for 5 min. SDS-PAGE was carried out on Bis-Tris gels, except when MITF-hyperphosphorylation was to be separated, the samples were run on 12.5% Tris-Glycine gels made with 200:1 acrylamide:bisacrylamide (custom order, Severn Biotech). Gels were transferred onto Immun-Blot Low Fluorescence PVDF Membrane (Biorad) at 100 V for 60 min for Western-blotting. Blotting involved blocking for 1 h, room temperature in 5% non-fat milk in TBS containing 0.25% TWEEN-20 (TBST). Antibody incubations were done in 5% BSA-TBST overnight, 4 °C for primary antibodies and 1 h RT for secondary antibodies. Visualization was carried out on X-ray film using ECL (GE) or ChemiDoc (Biorad).

**Immunoprecipitation.** All steps involved in affinity purifications were carried out at 4 °C using LoBind Tubes (Eppendorf#Z666505-100EA). Transiently transfected cells were trypsinised, washed in PBS and pelleted by centrifugation at 1200 × g for 4 min. Cell pellets were lysed in RIPA supplemented with 4× protease inhibitor cocktail (Roche# 11836145001). The cell suspensions were briefly sonicated for 30 s

(7.5 s on/off cycle) using Bioruptor® Plus (Diagenode) then cleared by centrifugation at 14000 × g for 10 min. The supernatants were transferred to a fresh tube, keeping 10% as input and the remainder was rotated overnight with 20 µg antibody. An appropriate amount (10% above the theoretical binding capacity) of Dynabeads Protein G (Invitrogen#10004D) were washed in lysis buffer and blocked overnight with 0.5 mg/ml BSA (Sigma #B8894) before adding to the samples and rotated for 1 h. The beads were washed 6-times with lysis buffer, transferring to a new tube after the first wash, and finally boiled in 4× LDS.

**GFP-tag affinity-purification.** GFP pull-down samples were purified using from Phoenix-ampho cell line transiently transfected (16 h) with 6 µg of the indicated plasmids in a 6 cm petri dish using FuGENE VI as described above. Cells for each experiment were lysed in 500 µl RIPA supplemented with cOmplete™ ULTRA protease inhibitor cocktail (Sigma Cat#5892988001) and 5 µM M344 (Stratech Scientific Cat#S2779-SEL) on ice for 30 min and subjected to mechanical disruption using 25 gauge needle till no visible clump can be seen. 100 µl of washed, pre-equilibrated GFP-trap (chromotek Cat#gtma-100) in lysis buffer was added to each precleared (14,000 × g for 10 min) sample, incubated overnight, 4 °C, washed thrice in RIPA without NP-40 and eluted in LDS, 95 °C for 10 min before SDS-PAGE.

**His-tag affinity-purification.** Cells were treated with 5 µM M344 (Tocris#2771) and 200 nM TPA (Sigma#4174 S) at the indicated time before lysis in 1 ml His-purification lysis buffer (6 M guanidinium HCl, 0.1 M Na$_2$HPO$_4$/NaH$_2$PO$_4$, 10 mM Tris-HCl (pH8), 0.005 M imidazole, 0.01 M $\beta$-ME, 1x EDTA-free protease inhibitor cocktail (Roche# 5892791001) for 5 min room-temperature before preclearing by centrifugation at 14000 × g for 10 min, 4 °C. Supernatant was then transferred to 15 ml Falcon tube (Sigma#CLS430052), topped up with 4 ml His-purification lysis buffer and rotated overnight, 4 °C in Ni-NTA resin (Merck# 70691-3). Ni-NTA resin was equilibrated in 2 times bead volume in His-purification lysis buffer for 5 min, room-temperature before adding to the lysate. The beads were washed once in His-purification lysis buffer and three times in His-purification wash buffer (8 M Urea, 0.1 M Na$_2$HPO$_4$/NaH$_2$PO$_4$, 10 mM Tris-HCl (pH8), 0.005 M imidazole, 0.01 M $\beta$ − ME) rotating 5 min between each wash. The washed-resins were directly eluted in LDS buffer.

**Peptide array.** Peptides of a total 14 amino acids were printed on cellulose membranes using SPOT technology[83] with at least 5 residues to the left or right of the indicated acetylated or non-acetylated residue. Membranes were activated in EtOH and rinse thoroughly with TBST prior to blotting. The array blotting was carried out as with Western-blotting, except the blocking was carried out for 4 h.

**Mass spectrometry and sample preparation.** HEK 293 cells were transfected with a mammalian Myc-epitope-tagged murine MITF expression vector together with expression vectors for either CBP or p300. 48 h post-transfection cells were harvested, and whole cell lysate was used for immunoprecipitation using anti-Myc antibody. Immunoprecipitates were resolved using SDS PAGE and gel bands corresponding to MITF were excised, destained then reduced and alkylated and digested with chymotrypsin overnight. Digests were analysed using a LTQ XL Orbitrap (Thermo, Hemel Hempstead), coupled to a Dionex Ultimate 3000 nano HPLC system (Camberley, Surrey). Peptides were resolved using a 75 micron internal diameter by 15 cm long C18 home-packed emitter column, packed with Dr maisch C18 material. The injection volume was 2 µL and the samples were resolved using a 40 min gradient running from 5% acetonitrile + 0.1% formic acid to 35% acetonitrile +0.1% formic acid. Data were analysed using Mascot (Matrixscience, London). Precursor mass tolerance was set to 10 ppm, fragment mass tolerance was 0.5 Da, fixed modification was

carbamidomethylation of cysteine and variable modification was oxidized methionine and acetylated lysine. Data were searched against an IPI mouse database.

**Recombinant protein purification.** MITF DNA-binding domains (residues 180–296) that were either WT, K206Q or K206R were expressed in *Escherichia coli* BL21(DE3) cells (Agilent Technologies). Cultures were grown in Luria-Bertani broth to an OD$_{600}$ of 0.7–0.8. Recombinant protein overexpression was induced by addition of isopropyl β-D-1-thiogalactopyranoside (IPTG; 0.5 mM final concentration) and the cultures were incubated for a further 6 h. Cells were harvested by centrifugation, washed in PBS and frozen on dry ice. After thawing, cells were suspended in lysis buffer (50 mM NaH$_2$PO$_4$, 300 mM NaCl, 10 mM imidazole, 10% v/v glycerol, pH 7.4 [NaOH], 20 mg/ml lysozyme [Invitrogen], 1× protease inhibitor cocktail [Roche]). Cells were lysed by sonication and centrifuged. Clarified lysate was mixed by rotation with a 50% Ni-NTA slurry (QIAGEN) previously equilibrated in lysis buffer and loaded in gravity flow columns (Bio-Rad). After extensive washing in wash buffer (50 mM NaH$_2$PO$_4$, 300 mM NaCl, 20 mM imidazole, 10% v/v glycerol, pH 7.4 [NaOH]), bound material was serially eluted in elution buffer (50 mM NaH$_2$PO$_4$, 300 mM NaCl, 20 mM imidazole, 10% v/v glycerol, pH 7.4 [NaOH]). Fractions were analysed by SDS-PAGE and Coomassie staining to determine purity, and pure fractions were pooled and glycerol added to a final concentration of 30% v/v.

**Recombinant Acetyl-K206 MITF**

cDNA encoding the helix-loop-helix region of the human MITF was fused by polymerase chain reaction (PCR) downstream to a Tobacco Etch Virus (TEV)-cleavable N-terminal hexa histidine affinity tag (extension: MHHHHHHSSGVDLGTENLYFQ*A, where A is the first amino acid of the MITF fragment) in presence of Herculase II fusion DNA polymerase (Agilent Technologies). An Amber codon (TAG) was introduced by PCR-based mutagenesis at K206 and the final PCR product subcloned in frame between the NcoI/XhoI sites of the pCDF-pYIT vector. Acetylation of the MITF protein at K206 was performed in vivo using amber stop codon/suppressor tRNA technology with several modifications that increased overall yields from 10% to above 70 % as previously described[84]. In brief, the Amber K206 mutant MITF was co-expressed in Tuner™ BL21 DE3 cells (Novagen, #70622) with an orthogonal N(epsilon)-acetyl-lysyl-tRNA synthetase/tRNA(CUA) pair, leading to site specific incorporation of N(epsilon)-acetyl-lysine. Cells were grown at 37 °C and at an OD$_{600}$ of ~0.7 the culture was supplemented with 20 mM nicotinamide (NAM, Sigma, #N3376-100G) and 25 mM acetyl-lysine (H-Lys-Ac-OH, GL Biochem Shangai Ltd, #GLS140325). Protein expression was induced 30 min later by addition of 50 µM IPTG and cells harvested by centrifugation (8700 × g, 15 min, 4 °C; Beckman Coulter Avanti J-20 XP centrifuge) after 6 h at 37 °C before re-suspension in lysis buffer [50 mM HEPES, pH 7.5 at 20 °C, 500 mM NaCl, 5% glycerol, 1 mM tris(2-carboxyethyl)phosphine (TCEP) and 1:1000 (v/v) Protease Inhibitor Cocktail III (Calbiochem)]. Cells were then lysed three times at 4 °C using a Basic Z Model Cell Disrupter (Constant Systems Ltd, UK) and DNA removed by precipitation on ice for 30 min with 0.15% (v/v) of polyethyleneimine (PEI). Lysate was cleared by centrifugation (16,000 × g for 1 h at 4 °C). The supernatant was applied to a Cobalt-based IMAC resin column (Talon, GE Healthcare, 5 mL, equilibrated with 20 mL lysis buffer) and the column washed once with 30 mL of lysis buffer, and then with 20 mL of lysis buffer containing 30 mM imidazole. Protein was next eluted using a step gradient of imidazole in lysis buffer (50, 100, 150, 2×250 mM imidazole in 50 mM HEPES, pH 7.5 at 25 °C, 500 mM NaCl and 5% glycerol). All fractions were collected and monitored by SDS-polyacrylamide gel electrophoresis (Bio-Rad Criterion Precast Gels, 4%–12% Bis-Tris, 1.0 mm, from Bio-Rad, CA.). Clean fractions were pooled and protein was treated overnight at 4 °C with TEV protease to

remove the hexa-histidine tag. Untagged protein was further purified by size exclusion chromatography on a Superdex 75 16/60 HiLoad gel filtration column (GE Healthcare Life Sciences) on an AktaPrime plus system (GE/Amersham Biosciences). Recombinant acetylated MITF eluted as single symmetrical monomeric peak. Eluted protein fractions were monitored by SDS-polyacrylamide gel electrophoresis and concentrated in gel filtration buffer (10 mM HEPES pH 7.5, 500 mM NaCl, and 5% glycerol) using Amicon Ultra (EMD Millipore) concentrators with a 3 kDa MW cut-off. Protein was aliquoted into 100 mL fractions, flash frozen in liquid nitrogen and stored at 80 °C until further use. Protein handling was performed on ice or in a cold room. In order to verify the degree of MITF acetylation, purified acetyl MITF was subjected to Electro-spray Quadrupole Time of Flight Mass Spectrometry according to the protocol described in Chromatography 2014, 1, 159–175; doi:10.3390/chromatography1040159. Purified protein (1 mg/mL) was diluted 1:60 in 0.1% (v/v) formic acid (60 µL final volume). Reverse-phase chromatography was then performed in-line prior to mass spectrometry using an Agilent 1100 HPLC system (Agilent Technologies Inc., Palo Alto, CA, USA). 50 µL was injected onto a 2.1 mm × 12.5 mm Zorbax 5µm 300SB-C3 guard column (Agilent Technologies, AG860950-909) housed within a column oven kept at 40 °C. The solvent system used consisted of 0.1% (v/v) formic acid in LC-MS grade water (Millipore, solvent A) and 0.1% (v/v) formic acid in methanol (LC-MS grade, Chromasolv, solvent B). Chromatography was performed as follows: Initial conditions were 90% buffer A and 10% buffer B with a flow rate of 1.0 mL/min. After 15 s at 10% buffer B, a linear gradient from 10% to 80% buffer B was applied over 45 s, followed by 80% to 95% buffer B over 3 s. Sample elution continued isocratically at 95% B for 72 s followed by equilibration at initial conditions for a further 45 s. Protein intact mass was determined using an MSD-TOF electrospray ionization orthogonal time-of-flight mass spectrometer (Agilent 6530 QTOF, Agilent Technologies Inc., Palo Alto, CA, USA). Raw ion count data deconvolution was carried out using the Mass Hunter WorkStation software (Qualitative Analysis Vs B.06.00 (Agilent Technologies, Palo Alto, CA). The correct intact mass within 1 Da was confirmed for the recombinant Kac MITF protein.

**Chromatin immunoprecipitation and ChIP-Seq.** For each biological replicate, cells from 30 80% confluent 15 cm dishes were trypsinised, collected into a 50 ml falcon tube (Corning; Cat# 430828), centrifuged (800 × g, 4 min) and media aspirated in a batch of 3 dishes before cross-linking in 45 ml ice-cold 0.4% paraformaldehyde PBS at room temperature with rotation for 10′. Quenching was carried out by adding glycine to a final concentration of 0.2 M and rotate for a further 10′ and collected by centrifugation (1500 × g, 10 min) and snap-freeze on dry-ice. Each cell pellet was lysed in 1 ml ChIP lysis buffer (50 mM Tris-HCl (pH 8.0), 10 mM EDTA, 10 mM sodium butyrate, 1% SDS, 4×PIC (Roche; Cat#05056489001)) and pass through a 25 guage needle until there were no visible clumps before sonicated for around 12 min in a Covaris S220 (Peak incident = 145 W, Duty Factor = 8%, Cycle/Burst = 200). Fragmentation was assessed on Bioanalyzer using Agilent High Sensitivity DNA Kit (Agilent#5067-4626) additional sonication was carried out until ~200–400 bp range was obtained. The sonicated chromatin was precleared prior to the ChIP by centrifugation at 13,000 × g, 10 min. The supernatant for each biological replicate was pooled and transferred to a 50 ml DNA-LoBind tubes (Eppendorf# EP0030122232) prior to 8-fold dilution in ChIP dilution buffer (16.7 mM Tris (pH 8.0), 167 mM NaCl, 1.2 mM EDTA, 1% Triton X-100, 0.01% SDS). 120 µg of anti-HA antibody (Roche; Cat# 11666606001) was used per biological replicate and rotated overnight. In parallel 550 µl Dynabeads protein-G were pre-equilibrated in 1 ml ChIP dilution buffer, washed and resuspended in 500 µl ChIP dilution buffer supplemented with 0.5 mg/ml BSA. Antibody capturing was done using blocked-Dynabeads for 1 h, 4 °C with rotation. Washing was performed in low salt wash buffer (20 mM Tris-HCl (pH 8.0), 150 mM NaCl, 2 mM EDTA,

1% Triton X-100, 0.1% SDS), high salt wash buffer (20 mM Tris-HCl (pH 8.0), 500 mM NaCl, 2 mM EDTA, 1% Triton X-100, 0.1% SDS) and LiCl wash buffer (10 mM Tris-HCl (pH 8.0), 250 mM LiCl, 1 mM EDTA, 1% sodium deoxycholate,1% NP-40) for 3 times in each buffer with beads transferred to a new DNA LoBind tube (Eppendorf; Cat# Z666548) after each wash. Elution was performed in 0.2 ml warm elution buffer (100 mM NaHCO₃, 1% SDS). Reverse cross-linking of ChIPed-DNA was done in two-step. First, addition of 0.3 M NaCl (final concentration) and 100 unit RNAse (Life Tech#AM2294) and incubated at 55 °C 2 h. Second, 18 µg Proteinase K (Sigma#3115828001) was added and incubated for a further 2 h. ChIPed-DNA was purified using QIAquick PCR Purification Kit (QIAGEN; Cat# 28106) concentration determined using Qubit dsDNA HS Assay Kit (Invitrogen; Cat# Q32851). Library preparation and 75 bp paired-end sequencing on an Illumina HiSeq 4000 at the Wellcome Trust Centre for Human Genetics (Oxford, UK).

**Nucleic Acid methods.** Total RNA was extracted cells using the RNeasy mini kit (QIAGEN#74104). cDNA was generated using Quanti-Tect Reverse Transcription Kit (QIAGEN#205313). PCR was carried out using KAPA HiFi HotStart ReadyMix (KAPABiosystem#KK2601).

**Bioinformatic analysis.** The two technical replicates (separate flow cell) were stitched together in UNIX to generate a single fastq. Raw fastq files were fastQCed, processed and mapped to human genome build hg38 (GRCh38v23) using STARv 2.5.1b[85]. Mapped SAM-files were used for peak calling using the Homer package[86]. Input from each sample were used as control for peak calling. Annotation, intersection, gene ontology, genome ontology, bedgraph and multiwig were performed with Homer. de novo motif identification was carried out using MEME. Exact peak size was used for all the analysis rather than arbitrary extending reads from the peak summit using average fragment size in each experiment. Histogram of read density were visualized using TreeView[87].

**Fluorescence anisotropy assay**
The following fluorescein-labeled oligonucleotides were synthesized at METABION (Planegg/Steinkirchen, Germany):

 CLEAR-box: 5′- GAGATCACGTGATGAC-3′-Fluorescein
 TYR: 5′- GAGATCATGTGATGAC-3′-Fluorescein
 M-box: 5′- GAGATCATGTGTTGA C −3′-Fluorescein

 These oligonucleotides were annealed with complementary unlabeled oligonucleotides through incubation at 95 °C for 5 min, followed by a passive cooling step to room temperature. Increasing concentrations of MITF proteins were incubated with the respective dsDNA oligonucleotides at a final concentration of 1.33 nM at 25 °C for 5 min in 10 mM Tris/HCl pH 7.5, 300 mM NaCl, 0.01% Triton X-100, and 0.1 mg/mL BSA. Fluorescence anisotropy was then measured using an Infinite M1000 plate reader (TECAN) using the excitation diode at 470 nm and detecting the emitted light at 530 nm.

$$y = B_0 + \frac{(B_{max} - B_0) \times (C + x + K_D - \sqrt{(C + x + K_D)^2 - 4Cx})}{2C} \quad (1)$$

 The anisotropy data collected using Tecan iControl version 1.10.4.0 were fitted using GraphPad Prism version 9.3.1 software to obtain dissociation constants based on the above quadratic equations where C is constant and corresponds to the oligonucleotide concentration, $B_{max}$ is the fitted maximal anisotropy value, $B_0$ is the fitted minimal anisotropy value, $x$ the concentration of protein, and y the anisotropy values.

 The KD values reported in Fig. 4c correspond to the means of three independent measurements and the +/−error numbers represent the standard deviations.

## Single molecule tracking

SMT analysis was performed in 4-well Nunc Labtek Chambers (Thermo-Fisher, Milan, Italy) on cells stably expressing doxycycline-inducible HALO-tagged MITF WT or K206 mutants treated for 24 h with 20 ng/ml of doxycycline. The HALO tag was labeled using 100 pM of the cell-permeable HaloTag ligand Janelia Fluor® 549 (Janelia Farm, Ashburn, VA, USA) to each well. After 30 min incubation cells were extensively washed with PBS, incubated for 20′ at 37 °C in culture medium, followed by one last wash with PBS and one in RPMI 1640. The acquisition of single molecule movies were performed according to two different regimes on a custom-built HiLo microscope, equipped with a stage temperature and $CO_2$ controller (OkoLab, Naples, IT) - set to 37 °C and 5% respectively −, a 561 laser (Cobolt 06-01 series, Hubner Photonics, DE) and an Hamamatsu Fusion sCMOS camera (Hamamatsu Photonics Italia SRL, Arese, IT). To quantify diffusion coefficients and bound fraction, movies were collected using stroboscopic illumination setting the laser exposure of 5 ms, the laser power density to approximately 0.8 kW/cm² and the acquisition frame rate to 100 fps. Movies were tracked using the ImageJ/Fiji plugin Trackmate[88], and tracks were then analysed using our previously described software developed in Matlab[50,54]. Briefly, the tracks are used to populate the histogram of displacements (bin size $\Delta r = 20$ nm), that are then fit with a three-component diffusion model that provides the probability of observing a displacement in the interval $\left[r - \frac{\Delta r}{2}, r + \frac{\Delta r}{2}\right]$ :

$$p(r) = r\Delta r \sum_{i=1}^{3} \frac{F_i}{2D_i\Delta t} \exp\left(-\frac{r^2}{4D_i\Delta t}\right) \qquad (2)$$

Where $\Delta t$ is the interval between two acquisitions (10 ms) and $F_i$ is the fraction of molecules with diffusion coefficient $D_i$. Fits are performed on the histograms of displacements obtained for each cell. The slowest diffusing component ($D_1 = 0.034$ μm²/s) is compatible with previously reported diffusion coefficients of chromatin bound proteins, so that $F_1$ represents the fraction of molecules detected as bound. Parameters resulting from the fit, the fractions $F_i$ and the diffusion coefficients $D_i$, are compared across the mutants by non-parametric Kruskal-Wallis test.

In order to measure the duration of MITF binding events (the residence times), we acquired movies at lower laser power (scaled down by a factor 20) and collected movies using long exposures (200 ms exposure, frame rate 2 Hz), in order to blur out diffusing molecules and selectively image immobilized ones[56]. These slow movies were analysed by kymograph analysis as described in[54], and the duration of the immobile track segments were used to populate the complement cumulative distribution of the duration of binding events, 1-$CDF$ – accounting for immobile molecules lasting longer than 5 frames (2.5 s). Disappearance of bound molecules can be due to either unbinding or photobleaching. To correct for photobleaching artefacts we adopted a recently described procedure[57]. Briefly, we collected the apparent distribution of residence times for HaloTag-H2B upon the same experimental conditions. The H2B distribution was then fit with a multi-exponential decay and the slowest component of such fit was used to provide an estimate of the photobleaching rate ($k_b = 0.0316 s^{-1}$). The corrected distributions of residence times show in Fig. 3E, were then obtained by dividing the uncorrected distributions for $\exp(-k_b t)$. Error bars are provided by generating multiple (1000) distributions of residence times, each after dropping 50% of the data. The (restricted) mean-survival time was obtained by integrating the 1-CDF curve between 2.5 s (shortest measured survival time) and 100 s (longest measured survival time).

## Reporting summary

Further information on research design is available in the Nature Portfolio Reporting Summary linked to this article.

## Data availability

The ChIP-seq data generated in this study are available in the Gene Expression Omnibus database under accession code GSE137776 and can be found here: https://www.ncbi.nlm.nih.gov/geo/query/acc.cgi?acc=GSE137776. Source data are provided in this paper. Source data for western blots are provided in Supplementary Fig. S6; for the fluorescent anisotropy data presented in Fig. 4, and for the zebrafish experiments shown in Fig. 5, source data are provided in the Supplementary Source Data file. Source data are provided in this paper.

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

## Acknowledgements

The Piggybac vectors were kindly provided by Kazuhiro Murakami (RIKEN, Kobe, Japan). This work was funded by the Ludwig Institute for Cancer Research (C.R.G.); Cancer Research UK (CRUK) grant number C38302/A12981, through a CRUK Oxford Centre Prize DPhil Studentship (H.F.); The Medical Research Council grants MR/N010051/1 (P.F.) and MC_UU_00007/9 (Z.Z., E.E.P.); L'Oreal-Melanoma Research Alliance 401181 (E.E.P.); The Wellcome Trust and The Postdoc Fund of the University of Iceland (A.S.); The Research Fund of Iceland (E.S.); a European Research Consolidator Award ZF-MEL-CHEMBIO 648489 (E.E.P.); NIH R01 CA268597-01 (C.R.G., PL); the Italian Foundation for Cancer Research AIRC IG-2018-ID21897 (D.M., A.Lo); A Fonds de Recherche du Québec-Santé Junior 2 salary award (J.-P.L.) and a Cancer Research Society operating grant (935296) to J.-P.L. (A.La).

## Author contributions

Conceptualization, C.R.G. Supervision, C.R.G., E.S., M.W., E.E.P., P.F., D.M., J.-P.L., Visualization, P.L., S.P., A.S., Z.Z., Formal Analysis, P.L., A.Lo, V.P., Z.Z., B.T., J.-P.L. Investigation, P.L., A.Lo, H.F., A.S., Z.Z., V.P., S.P., B.T., A.La. Writing-Original draft, C.R.G., P.L. Writing-review and Editing, C.R.G., P.L., D.M., J.-P.L. Funding acquisition, C.R.G., E.S., E.E.P., P.F., M.W., D.M., J.-P.L.

## Competing interests

The authors declare no competing interests
