## [Peer Review File · Nature Communications]

Acetylation reprograms MITF target selectivity and residence timeREVIEWER COMMENTS

Reviewer #1 (Remarks to the Author):

This is a very interesting and generally well-performed study. Target selection for transcription factors is an area of high interest and importance. The methodology and logic of the study employed here is an ideal way to shed light on these processes and dissect how TFs achieve this activity.

In particular, the authors have performed a multi-disciplinary study to determine the impact of acetylation upon the activity of MITF. Overall, this provides a robust model/mechanism for how MITF differentiates between DNA targets during differentiation and proliferation.

I have the following comments on the work which should be addressed before publication:

- (1) Can you provide quantification for the data relating to the specificity of the anti-acetyl K206 antibody. The language in the manuscript is a little loose/non-specific (for example, "only weakly binding").
- (2) For Fig2 (SPT) – what are the relative expression levels for the constructs? It is important to understand the levels because the current analysis breaks the measurements down in to fractions of molecules in each state. Different expression levels may lead to saturation of certain states thereby giving artificial increases in others.
- (3) For Fig2 – can endogenous protein cause disruption to binding states? Can you knockdown endogenous then perform these measurements?
- (4) For Fig2 – Did you perform a control measurement with an isolated Halo domain? It is a useful control to determine a baseline (non-specific) "static" fraction.
- (5) For Fig2- Please you plot the dissociation rate (lifetime) vs frequency for each molecule to give an overview of all the data rather than fractions.
- (6) For Fig2 – you performed experiments across 28 cells but from how many experiments?
- (7) For Fig3 – There are some significant problems with the analysis for the Fluorescence anisotropy experiments. Firstly, the binding model used cannot be applied to the data, as performed in the experiment. This model only works when the protein concentration is at least 10x higher than the DNA concentration. You should therefore fit the data using a quadratic binding curve.
- (8) For Fig3 - Anisotropy is strongly influenced by fluorescence change. For example, protein-DNA can have a different fluorescence intensity compared to DNA alone. This factor directly ties in to correctly calculating the Kd. This is very important when using mutant proteins which may lead to different fluorescence change upon binding to the DNA. You can therefore use a specific quadratic binding curve to fit the anisotropy data – See Fili et al Nature communications 2017 8 1871 for an example on how to fit the data to these equations.
- (9) For Fig3, anisotropy is an absolute value there is no need to display data on a normalised scale (like you would for fluorescence intensity). It is vital to display the true values so that the free DNA and protein-DNA complexes are shown at expected/sensible levels which can be estimated on molecular weight.

Reviewer #2 (Remarks to the Author):

This is an interesting study by Louphrasitthiphol and colleagues describing the effects of acetylation on MITF and its implications to Waardenburg syndrome. The work expands upon the author's previous 2020 study investigating the use of acetylation to alter transcription factors binding affinities.

Concerns:

- 1.) Within your mass spectrometry and sample preparation section there is not enough information for a reader to replicate your experiment. Authors should report the injection volume of samples, column used, and gradient used to obtain results.
- 2.) Within the Recombinant Acetyl-K206 MITF section it is stated "purified protein was diluted to 1 mg/mL with 0.1% formic acid and 60 mL injected on an Agilent 6530 QTOF." Typical proteomics analysis utilizes 200 ng of protein per sample injection of approximately 1-5 μ L. Please double check the work reported. For reproducibility, please also include the gradient utilized for analysis or state if it was a direct injection into the instrument.
- 3.) The reported Zorbax column "Zorbax 5 mm 300SB-C3 column" should also be confirmed. I am unsure as to whether the 5 mm refers to a rounded-up value of inner diameter as there is not currently a column of that length seen on the Agilent website.
- 4.) The denoted peptide sequences in supplemental figure 1 appear as though they were digested using trypsin and not chymotrypsin as stated in the article.

Overall, the paper was very thorough in using complimentary techniques to investigate changes occurring to binding affinity due to acetylation. I look forward to reading more work from this lab in the future.

Reviewer #3 (Remarks to the Author):

This manuscript by Louphrasitthiphol and colleagues analyzes how post-translation modification of the transcription factor, MITF, alters its chromatin binding activity. The authors show that acetylation of MITF at the K206 residue selectively reduces the ability of MITF to bind to its targets and regulates melanocyte differentiation in a zebrafish model. Due to the central role of MITF in phenotypic switching, drug tolerance and minimal residual disease cell states in melanoma, this study has the potential to be highly relevant and novel. Furthermore, the authors generate novel reagents such as antibodies to detect acetylated MITF and also use of HALO-tagged version of MITF chromatin binding mobility assays. Indeed, while the authors show pretty rigorously that MITF acetylation at K206 reduces the binding of MITF to some of its target genes, especially those associated with differentiation, the study lacks evidence that the expression of the mentioned genes is actually reduced. Some other concerns listed below regard the rigor of the experiments. We believe that in its current form, the manuscript is novel and very promising. If the authors could address/clarify our concerns, it would be suitable for publication.

1. Figure 1g, why was an inducible model for MITF utilized. I understand the rationale for p300 but not for inducible regulation of both transgenes.
2. Mechanistically, how MEK signaling regulates the acetylation of MITF is unclear. Is there interplay between ERK1/2 phosphorylation of MITF and its acetylation? Or is p300 activity generally reduced following MEK inhibition. The study would also be strengthened by more analysis of endogenous MITF. For example, in Figure 1h, do MEK inhibitors alter the acetylation of endogenous MITF?
3. The data in Figure 5 would be strengthened if extended to RNA-seq transcriptome analysis.
4. A general concern is that most of the experiments are performed in a single cell line. Despite the biochemical nature of the manuscript, key experiments should be performed in two or more melanoma cell lines.

Minor comments

1. Across the manuscript, please clarify in the figure legend and/or in the figures which cell lines have been used and for how long they were treated.
2. Figure 1f- please give the number of repeat experiments.
3. Please add statistical analysis to fig 2d and 3a.
4. In the Fig 1b legend, please clarify the additional sites that are shown at the top of the schematic.
5. In Fig 1g, please include a blot to demonstrate inducible expression of p300.
6. Please indicate the n used for each condition in Fig 4c in the figure legend and methods.
7. Clarify what is meant by R1 and R2 in the Fig 5c legend.

RESPONSE TO REVIEWER COMMENTS

Reviewer #1 (Remarks to the Author):

This is a *very interesting and generally well-performed study*. Target selection for transcription factors is an area of high interest and importance. The *methodology and logic of the study employed here is an ideal way to shed light on these processes* and dissect how TFs achieve this activity.

In particular, the authors have performed a multi-disciplinary study to determine the impact of acetylation upon the activity of MITF. Overall, this *provides a robust model/mechanism for how MITF differentiates between DNA targets during differentiation and proliferation*.

I have the following comments on the work which should be addressed before publication:

(1) Can you provide quantification for the data relating to the specificity of the anti-acetyl K206 antibody. The language in the manuscript is a little loose/non-specific (for example, “only weakly binding”).

We have now added a quantification to the dot blot in Fig. 1e, and modified the language used to describe the specificity of the antibody. Note that the differential recognition between Ac-K206 and the Ac-K33 is likely to be much greater than the quantification suggests since the signal at K206 is saturated.

(2) For Fig. 2 (SPT) – what are the relative expression levels for the constructs? It is important to understand the levels because the current analysis breaks the measurements down in to fractions of molecules in each state. Different expression levels may lead to saturation of certain states thereby giving artificial increases in others.

We agree with the referee on this point, and the referee raises a simple question that took us a very long time to address to our satisfaction.

The SMT experiments were designed so that the HALO-Tagged MITF is inducible by doxycycline. Although each cell line made may express different levels of ectopic MITF at the same concentration of doxycycline, by carefully titrating the doxycycline levels we are able to achieve similar expression levels in cells expressing WT or mutant MITF (we have previously ascertained that changing Dox levels alone does not affect the SMT results).

In the initial submission we titrated the levels of Dox to understand how much was required to generate similar levels of MITF. We subsequently shipped the cells to our collaborators in Milan who undertook the SMT experiments using the same levels of Dox (20 ng/ml) and prepared the figure in the original submission. However, following the referee’s comment, the cell lines were then subjected to a new Western blot in Milan that revealed differences in HALO-MITF expression, presumably arising at some stage following the cell lines being placed into culture in Milan.

We therefore checked whether using 0, 20, and 100 ng of doxycycline to express MITF to different levels would affect to the outcome of the SMT experiments (Note that 20 ng equates to physiological levels of MITF). The results (Figure 1 for referee) indicated that changing the levels of Dox from 20 ng to 100 ng (Fig. 1a) that led to around a 5-fold increase in HALO-MITF protein would have a very minor effect on the bound, slow and fast diffusing fractions (Fig. 1b). However, the differences we noted in the fast and slow diffusing fractions were not as great as those we observed between the MITF K206R and Q mutants (Fig. 1c) that in this experiment exhibited only a small difference in expression. Overall these experiments suggest that the contribution of different MITF levels, at least within the range used in in the SMT experiments, would make only a minor contribution to any differences seen between MITF WT and mutants.

Nevertheless, we then prepared further cell lines and spent a long time carefully adjusting the Dox concentrations until we achieved similar levels of WT and mutant MITF at the same time as the SMT experiments were performed. This new WB is now shown in new Fig. 2a.

We then repeated the SMT experiments. The updated Figure 2, now displays these new experiments obtained with similar expression levels of Halo-MITF (Fig. 2b), so that we can accurately compare results between the WT and mutants. Our previous results were recapitulated in these new cell lines: Our new data confirms the observations in the previous version of the manuscript that WT MITF is tightly bound to chromatin with a large bound fraction, and long residence times exceeding 100s for about 40% of the molecules. Our new data also confirms that the acetylation mimicking MITF K206Q mutant binds chromatin less tightly than WT MITF or the K206R mutant

(3) For Fig 2 – can endogenous protein cause disruption to binding states? Can you knockdown endogenous then perform these measurements?

The referee is correct that endogenous MITF might affect the relative binding by the ectopic HALO-MITF WT and mutants, but a knockdown experiment would be complicated by the fact that efficacy of knockdown in individual cells might vary. To address this issue we have therefore repeated the SMT experiments originally performed in the MITF^{High} 501mel cell line in an undifferentiated melanoma line, IGR39, in which endogenous MITF is expressed poorly if at all. The results using WT and mutant MITF expressed to similar levels suggest that the fraction of MITF WT versus mutants in the bound, fast-diffusing, and slow-diffusing states is not dependent on the expression of endogenous MITF, and as in the 501mel cells, the bound fraction and the residence times using the K206Q mutant are reduced compared to WT or the K206R mutant. This new data is now displayed in new Supplementary Fig. S3.

(4) For Fig 2 – Did you perform a control measurement with an isolated Halo domain? It is a useful control to determine a baseline (non-specific) “static” fraction.

Yes, this was done and the control data is now included in Fig. 2c-d. As expected, the result shows that the HALO alone baseline exhibited a very low bound fraction and faster diffusion coefficients than MITF-WT or mutants.

(5) For Fig 2- Please you plot the dissociation rate (lifetime) vs frequency for each molecule to give an overview of all the data rather than fractions.

Indeed, Fig. 2e represents the 1 - cumulative frequency of the observed lifetimes (i.e. the survival time distribution), which is the standard representation of distribution of residence times in single molecule tracking studies. This is preferable than the probability density function (frequency of each measured lifetime), since for long-lived molecules we cannot provide an exact estimate of their residence time (due to photobleaching and to the limited duration of the movie), but just say that they are ‘longer than’ a certain time t .

In addition, we now provide an estimate of the mean (restricted) survival time, to provide a single parameter that can be used for comparison among different proteins. This data is presented for the 501mel cells in Fig. 2g that shows the bound fraction dissociation rate of WT vs mutant MITF and a comparison to both p53 and USF1 as control transcription factors. Note that USF1 can bind MITF binding sites but has a reduced bound lifetime in comparison to WT MITF.

(6) For Fig 2 – you performed experiments across 28 cells but from how many experiments?

Every experiment in Fig. 2 has been repeated twice after adjusting doxycycline levels. The number of experiments is now shown in the relevant figure legends. The experiments shown are just a fraction of the total SMT experiments performed while adjusting doxycycline levels. In any experiment when the protein levels were the same, the K206Q mutant displayed reduced bound fraction/residence time compared to WT or the K206R mutant.

(7) For Fig3 – There are some significant problems with the analysis for the Fluorescence anisotropy experiments. Firstly, the binding model used cannot be applied to the data, as performed in the experiment. This model only works when the protein concentration is at least 10x higher than the DNA concentration. You should therefore fit the data using a quadratic binding curve.

We would like to thank the reviewer for the given opportunity to reconsider the analysis of our fluorescence anisotropy data in a more rigorous manner. We indeed worked in a particular concentration regime which is at the borderline to allow hyperbolic fitting. As a consequence the K_d s corresponding to the tightest interactions (WT and K206R for the CLEAR and TYR motifs) were slightly overestimated, but were still within the given error range.

However, given the referee's comments we have now reprocessed all the data using the quadratic equation corresponding to our model. The Material and Methods section and Fig. 3 have therefore been modified accordingly. That said, we would like to point out that the reprocessing of the data did not affect any of the conclusions we had drawn previously.

(8) For Fig. 3 - Anisotropy is strongly influenced by fluorescence change. For example, protein-DNA can have a different fluorescence intensity compared to DNA alone. This factor directly ties in to correctly calculating the K_d . This is very important when using mutant proteins which may lead to different fluorescence change upon binding to the DNA. You can therefore use a specific quadratic binding curve to fit the anisotropy data – See Fili et al Nature communications 2017 8 1871 for an example on how to fit the data to these equations.

We agree with the reviewer that anisotropy is strongly influenced by fluorescence change. Nevertheless, in our setup, protein binding does not have any impact on the fluorescence property of the fluorophore. Indeed, the total intensity, monitored all over the titrations, did not change significantly upon binding of the protein. Therefore, no correction of the anisotropy value has been applied here. We have now specified this in the methods section. Total intensity data can be provided on request if necessary.

(9) For Fig 3, anisotropy is an absolute value there is no need to display data on a normalised scale (like you would for fluorescence intensity). It is vital to display the true values so that the free DNA and protein-DNA complexes are shown at expected/sensible levels which can be estimated on molecular weight.

We agree that there is no need to display data on a normalised scale. Nevertheless the main parameter of interest in such a titration experiment is the anisotropy change which directly correlates to the complex formation. Therefore, we present now the figures with the difference of anisotropy on the y-axes instead of the fraction bound value. Nevertheless, we argue that, in contrast to the graphic representation used in Fili et al, Nature Communications (2017) 8, 1871, the presentation on a logarithmic scale is more appropriate, first of all, to convey the message (clearly and scalable distinction between the high and low binders) and, above all, to judge the quality of the data (to check e.g. if the saturation is reached at the end of the titration, or if the binding is influenced by non-specific binding or if there is a potential binding cooperativity).

Reviewer #2 (Remarks to the Author):

This is an interesting study by Louphrasitthiphol and colleagues describing the effects of acetylation on MITF and its implications to Waardenburg syndrome. The work expands upon the author's previous 2020 study investigating the use of acetylation to alter transcription factors binding affinities.

Concerns:

1.) Within your mass spectrometry and sample preparation section there is not enough information for a reader to replicate your experiment. Authors should report the injection volume of samples, column used, and gradient used to obtain results.

This information has now been added to the methods section.

2.) Within the Recombinant Acetyl-K206 MITF section it is stated "purified protein was diluted to 1 mg/mL with 0.1% formic acid and 60 mL injected on an Agilent 6530 QTOF." Typical proteomics analysis utilizes 200 ng of protein per sample injection of approximately 1-5 μ L. Please double check the work reported.

1 mg /ml is correct, but the 60 ml should have been 60 μ L as the referee surmised. We thank the referee for spotting this error which has been corrected.

For reproducibility, please also include the gradient utilized for analysis or state if it was a direct injection into the instrument.

See response to point 3 below

3.) The reported Zorbax column “Zorbax 5 mm 300SB-C3 column” should also be confirmed. I am unsure as to whether the 5 mm refers to a rounded-up value of inner diameter as there is not currently a column of that length seen on the Agilent website.

The methods section relating to this part of the work has now been checked and the details of the gradient used added and more details provided.

4.) The denoted peptide sequences in supplemental figure 1 appear as though they were digested using trypsin and not chymotrypsin as stated in the article. The referee is correct and we have modified the text accordingly. We performed both trypsin and chymotrypsin in different experiments and mislabelled this one.

Overall, the paper was *very thorough in using complimentary techniques to investigate changes occurring to binding affinity due to acetylation*. I look forward to reading more work from this lab in the future.

Reviewer #3 (Remarks to the Author):

This manuscript by Louphrasitthiphol and colleagues analyzes how post-translation modification of the transcription factor, MITF, alters its chromatin binding activity. The authors show that acetylation of MITF at the K206 residue selectively reduces the ability of MITF to bind to its targets and regulates melanocyte differentiation in a zebrafish model. Due to the central role of MITF in phenotypic switching, drug tolerance and minimal residual disease cell states in melanoma, this study has the potential to be highly relevant and novel. Furthermore, the authors generate novel reagents such as antibodies to detect acetylated MITF and also use of HALO-tagged version of MITF chromatin binding mobility assays. Indeed, while the authors show pretty rigorously that MITF acetylation at K206 reduces the binding of MITF to some of its target genes, especially those associated with differentiation, the study lacks evidence that the expression of the mentioned genes is actually reduced.

Some other concerns listed below regard the rigor of the experiments. We believe that in its current form, *the manuscript is novel and very promising*. If the authors could address/clarify our concerns, it would be suitable for publication.

1. Figure 1g, why was an inducible model for MITF utilized. I understand the rationale for p300 but not for inducible regulation of both transgenes.

One of the issues we came across during the course of this study is that to get enough soluble MITF to be immunoprecipitated we need to overexpress it compared to the endogenous protein. However, by increasing the expression of MITF we can also titrate out its acetylation by endogenous p300. Since we used an inducible MITF elsewhere in the manuscript, we wanted to restore the balance of MITF and p300 activity by also expressing an inducible p300. We have now modified the text to make this clear and added a panel (Fig. 1g) to show the inducible expression of both MITF and p300.

2. Mechanistically, how MEK signaling regulates the acetylation of MITF is unclear. Is there interplay between ERK1/2 phosphorylation of MITF and its acetylation? Or is p300 activity generally reduced following MEK inhibition.

There were two likely ways in which MAPK signalling through MEK could affect p300-mediated MITF acetylation. First, MITF is phosphorylated by ERK on S73 and downstream by RSK on S409. It has been reported (Price et al) that phosphorylation of MITF at S73 leads to increased p300 binding. However, neither our lab nor that of Robert Ballotti were able to reproduce this phosphorylation-dependent interaction with p300 and we have shown that immunoprecipitation of CBP/p300 brings down both the S73 phosphorylated and non-phosphorylated forms of MITF. Moreover, depending on the gel used, MITF in western blots runs as two bands, with the upper band resulting from S73 phosphorylation. However, when we examine MITF acetylation using an anti-acetyl-Lysine antibody, both the upper and lower bands are acetylated (see new Fig. 1k for example), indicating that S73 phosphorylation is not required for MITF acetylation. This conclusion was confirmed using non-phosphorylatable S73A,S409A or double mutants that showed that mutant MITF was acetylated. These data suggest that modification of MITF by MAPK signalling is unlikely to play a role in p300 mediated MITF acetylation.

On the other hand, published data from Chen et al (2007) have revealed convincingly that phosphorylation of p300 by ERK increases its activity, and p300 activity is decreased by MEK inhibition. In agreement, we have published previously that BRAF, upstream from MEK/ERK also promotes a global increase in p300-dependent acetylation (Louphrasitthipol et al 2020). We have added the new Fig. 1k and modified the text to highlight these points.

The study would also be strengthened by more analysis of endogenous MITF. For example, in Figure 1h, do MEK inhibitors alter the acetylation of endogenous MITF?

As we have noted previously, working with endogenous MITF is notoriously difficult owing to the fact that it is very poorly extractable from nuclei (see Louphrasitthipol et al 2020). This is likely related to the observation that MITF has such a very long residence time as determined by SMT analysis compared to other transcription factors (residence time >100s with the dissociation curve flat at this time as shown in Fig. 2e). nevertheless we have shown that endogenous MITF is acetylated in an immunoprecipitation experiment presented in Fig. 1i.

The referee also suggests using MEK inhibitors to demonstrate a change in MITF acetylation. We have tried this experiment but found that that reproducibly MEK inhibitors led to loss of endogenous MITF expression. We therefore reply on the new experiments presented in Fig. 1k to demonstrate, as discussed above, using MITF mutants, that MAPK-mediated phosphorylation of MITF does not affect its global acetylation.

3. The data in Figure 5 would be strengthened if extended to RNA-seq transcriptome analysis.

We agree with the referee that ideally an RNA-seq analysis after the expression of WT MITF versus mutant MITF would add to the paper. However, we have found that inducing ectopic WT MITF expression does not yield substantial changes in gene expression, for example induction of known endogenous target genes. This is something that is known in the field, (see Vachtenheim et al J. Invest. Dermatol. (2001) 117,1505-1511; Gaggiolo et al Pigment Cell Res (2003) 16, 374-82 for examples), and that MITF expression in MITF-negative cell lines such as IGR39 cells does not lead to activation of differentiation target genes owing to their chromatinization. In another manuscript that we are in the process of preparing, we have observed that MITF appears both to activate or repress different sets of target genes under different conditions; that is, MITF would activate differentiation genes and repress proliferation genes during differentiation, but do the opposite in proliferating conditions. We suspect that post-translational modifications of MITF that are restricted to proliferation versus differentiation mediate changes in cofactor interaction that then dictate which genes are activated or repressed. In addition, we have also observed that MITF ability to regulate specific genes can be complemented by the MITF related factors TFEB and TFE3 that are also expressed in melanoma cells. Thus, our current experience is that gene expression studies using inducible MITF are at present difficult to interpret.

4. A general concern is that most of the experiments are performed in a single cell line. Despite the biochemical nature of the manuscript, key experiments should be performed in two or more melanoma cell lines.

The critical experiments here are those using single molecule tracking to determine in live cells the bound, fast-diffusing and slow-diffusing fractions of WT and mutant MITF. In response to Referee 2 we have now repeated these experiments in an additional cell line, IGR39, which express very low or no endogenous MITF (see for example Louphrasitthipol et al 2019). The results of these assays are now presented in new Supplementary Fig. S3 and show that the results in each cell line are broadly similar. In other words in live cells WT MITF or the K206R mutant binds much better than the K206Q acetylation-mimetic mutant irrespective of whether the cells used for the assay express endogenous MITF.

Minor comments

1. Across the manuscript, please clarify in the figure legend and/or in the figures which cell lines have been used and for how long they were treated.

We have modified the legend as requested.

2. Figure 1f- please give the number of repeat experiments.

This experiment has been performed once in 501mel cells as presented, but the effect of the p300/CBP inhibitor on MITF acetylation has also been determined in a different cell line (IGR37) and in 501mel cells

using a different MITF expression vector. We show just one example here. Because each 'repeat' was slightly different in experimental design, we have added a short description to the legend without indicating the 'n'.

3. Please add statistical analysis to fig 2d and 3a.

For Figure 2d we now have performed the analysis at the single cell level that allows us to appreciate the cell-to-cell variability in MITF dynamics and to perform statistics. A non-parametric KS test was performed to identify significant differences. P-values are now reported on the figures.

For Figure 3, it is very unusual to add a statistical test to a fluorescence anisotropy expt, but nevertheless we have performed unpaired two-sided t-tests and statistically significant differences ($P < 0.05$) are now designated with a * sign on Figs 3a and b.

4. In the Fig 1b legend, please clarify the additional sites that are shown at the top of the schematic
We have modified the legend as requested.

5. In Fig 1g, please include a blot to demonstrate inducible expression of p300.

We have now added (new Fig. 1g) a western blot to show that p300 and MITF are inducible using 50 ng doxycycline for 16 h. However, the level of induction of p300 at this concentration of Dox and at this time is low, though induction of p300 is clearly visible at a longer time point (72 h). That said, it appears that even a low level of induction of p300 is sufficient to induce acetylation of MITF, likely reflecting that p300 is highly active enzymatically.

6. Please indicate the n used for each condition in Fig 4c in the figure legend and methods.

Three independent zebrafish experiments were performed. We have modified the legend and methods accordingly.

7. Clarify what is meant by R1 and R2 in the Fig 5c legend.

R1 and R2 indicate two independent replicate ChIP expts. We have modified the legend as requested.

REVIEWERS' COMMENTS

Reviewer #1 (Remarks to the Author):

The authors have addressed my questions and comments. I thank the authors for their dedication to address these points. It is not easy to address controls with single molecule measurements so it is impressive to see the amount of work.

Reviewer #3 (Remarks to the Author):

The revised manuscript by Louphrasitthiphol and colleagues analyzes via a multi-disciplinary approach how post-translation modification of the transcription factor, MITF, alters its chromatin binding activity. This is a very clearly written manuscript on a topic that has particular importance to the melanocyte-melanoma field as well as more broadly to transcriptional regulation. New data in Figure 1i on endogenous MITF provide more rigor to the studies. The data on the K206Q mutant provide important insights to Waardenburg syndrome. These studies will inform work on other lineage transcription factors in the melanoma field. Clarifications, especially in the figure legends have been provided. Overall, this is an important and detailed manuscript that is worthy of publication in Nature Communications.

RESPONSE TO REVIEWERS' COMMENTS

We thank the referees for their positive assessment of the manuscript and their enthusiasm for the data presented.

We note that no further experimentation or modifications to the text are required by either referee.

Reviewer #1 (Remarks to the Author):

The authors have addressed my questions and comments. I thank the authors for their dedication to address these points. It is not easy to address controls with single molecule measurements so it is impressive to see the amount of work.

Reviewer #3 (Remarks to the Author):

The revised manuscript by Louphrasitthiphol and colleagues analyzes via a multi-disciplinary approach how post-translation modification of the transcription factor, MITF, alters its chromatin binding activity. This is a very clearly written manuscript on a topic that has particular importance to the melanocyte-melanoma field as well as more broadly to transcriptional regulation. New data in Figure 1i on endogenous MITF provide more rigor to the studies. The data on the K206Q mutant provide important insights to Waardenburg syndrome. These studies will inform work on other lineage transcription factors in the melanoma field. Clarifications, especially in the figure legends have been provided. Overall, this is an important and detailed manuscript that is worthy of publication in Nature Communications.